# Spatially resolved mapping of proteome turnover dynamics with subcellular precision

Feng Yuan[1,4], Yi Li[2,4], Xinyue Zhou[1], Peiyuan Meng[2] & Peng Zou [1,2,3] ✉

Cellular activities are commonly associated with dynamic proteomic changes at the subcellular level. Although several techniques are available to quantify whole-cell protein turnover dynamics, such measurements often lack sufficient spatial resolution at the subcellular level. Herein, we report the development of prox-SILAC method that combines proximity-dependent protein labeling (APEX2/HRP) with metabolic incorporation of stable isotopes (pulse-SILAC) to map newly synthesized proteins with subcellular spatial resolution. We apply prox-SILAC to investigate proteome dynamics in the mitochondrial matrix and the endoplasmic reticulum (ER) lumen. Our analysis reveals a highly heterogeneous distribution in protein turnover dynamics within macromolecular machineries such as the mitochondrial ribosome and respiratory complexes I-V, thus shedding light on their mechanism of hierarchical assembly. Furthermore, we investigate the dynamic changes of ER proteome when cells are challenged with stress or undergoing stimulated differentiation, identifying subsets of proteins with unique patterns of turnover dynamics, which may play key regulatory roles in alleviating stress or promoting differentiation. We envision that prox-SILAC could be broadly applied to profile protein turnover at various subcellular compartments, under both physiological and pathological conditions.

Eukaryotic cells are highly compartmentalized. To achieve proper biological functions, proteins are constantly trafficking within the elaborate cellular architecture throughout their life cycle. Take the secretory pathway as an example. Membrane receptors are initially synthesized in the endoplasmic reticulum (ER)[1] and subsequently transported to the Golgi apparatus before reaching the plasma membrane[2], where they replace "older" receptors that are sorted to the endosomes and destined for degradation in the lysosome[3] or the proteasome[4]. This journey covers nearly half of the cellular space. In each of these subcellular compartments, protein turnover dynamics is carefully orchestrated to ensure the proper folding and functioning of these biomolecules. The rate of turnover may vary substantially between different proteins[5–7], in different compartments, and may change rapidly when cells are under stress or undergo differentiation[8]. Thus, mapping protein turnover with subcellular precision is valuable for understanding cellular activities under physiological and pathological conditions.

A number of chemical tools have been developed to measure protein turnover dynamics at the whole-cell level. Newly synthesized proteins could be tagged with non-canonical amino acids[9,10]. For example, the application of bioorthogonal non-canonical amino acid tagging (BONCAT)[11] to hippocampal synapses discovered around 300

[1]Academy for Advanced Interdisciplinary Studies, PKU-Tsinghua Center for Life Science, Peking University, Beijing 100871, China. [2]College of Chemistry and Molecular Engineering, Synthetic and Functional Biomolecules Center, Beijing National Laboratory for Molecular Sciences, PKU-IDG/McGovern Institute for Brain Research, Key Laboratory of Bioorganic Chemistry and Molecular Engineering of Ministry of Education, Peking University, Beijing 100871, China. [3]Chinese Institute for Brain Research (CIBR), Beijing 102206, China. [4]These authors contributed equally: Feng Yuan, Yi Li. ✉e-mail: zoupeng@pku.edu.cn

differentially regulated proteins involved in biological processes such as neurite outgrowth and axonal guidance[11]. More recently, stable isotope labels (pulse-SILAC[12,13]) have been used to quantify protein turnover in mammalian cell lines, primary neuronal culture[6] and in vivo systems[14,15], revealing broad distributions in protein turnover half-lives in NIH3T3 cells (median 46 h)[16] and HeLa cells (mean 20 h)[13]. Applications of pulse-SILAC has also resolved the assembly kinetics of mitochondrial respiratory complexes[17], the mitochondrial ribosome[18], the nuclear pore complex[7], etc. While the above tools are powerful for analyzing protein turnover, they are only applicable to the whole-cell proteome level and lack subcellular spatial resolution.

To address this issue, pulse-SILAC labeling has been combined with organelle purification workflow, to specifically measure protein turnover dynamics in the mitochondria[17] and the nucleus[19]. However, organelle purification methods are prone to introducing contamination and are not generally applicable to other subcellular locations, which motivates us to explore alternative methods for studying subcellular proteome turnover. Over the past decade, enzyme-mediated proximity labeling techniques, including engineered promiscuous biotin ligases (BioID[20], TurboID[21]) and peroxidases (APEX2[22,23]/HRP[24]), have been developed for spatially resolved profiling of subcellular proteome. These genetically targetable enzymes are capable of generating highly reactive intermediates in situ (e.g. biotinyl 5′-adenylate[20,21] or biotin-conjugated phenoxyl free radicals[22–24]), which rapidly react with proximal proteins to form covalent bonds. However, TurboID labeling typically requires over 10 min, which is unsuitable for studying protein turnover dynamics. The high reactivity and short lifetime (<1 μs) of phenoxyl free radical have enabled APEX2/HRP to achieve 10 nanometer-scale spatial resolution within 1 min of labeling[25]. While these methods have been broadly applied to profile the abundance of proteins within subcellular structures (e.g. mitochondria[26,27], ER[28], primary cilia[29], etc.), they have not been used for investigating the turnover dynamics of subcellular proteome[25]. In a previous work, APEX2 labeling was coupled with multi-isotope imaging mass spectrometry to reveal the heterogeneity in protein turnover in lysosomes[30]. However, this work has not been extended to the proteome level.

In the current study, we aim to develop such a method, named prox-SILAC, which combines proximity labeling with pulse-SILAC metabolic tagging of nascent proteome. We choose APEX2/HRP as the proximity labeling method for its superior reaction kinetics. As outlined in Fig. 1A, APEX2/HRP is targeted to a specific subcellular location via fusion with signal sequence or protein markers. To quantify protein turnover dynamics, the cell culture medium is replaced with heavy SILAC medium containing isotope-labeled lysine and arginine for several hours[31,32]. Proximity labeling is triggered at the end of pulse-SILAC to label all proteins, both old and new, that are in the vicinity of APEX2/HRP. Biotinylated proteins are subsequently enriched and analyzed by liquid chromatography-tandem mass spectrometry (LC-MS/MS). The H/L ratios of mass spec intensities represent the metabolic replacement fractions (MRFs), which are used to quantify protein turnover.

## Results

### Characterizing prox-SILAC method in the mitochondrial matrix

We chose the mitochondria as a model to examine the spatial specificity and quantitation precision of prox-SILAC method. In human embryonic kidney 293 T (HEK293T) cells, APEX2 is targeted to the mitochondrial matrix (mito-APEX2) via N-terminal fusion with the mitochondrial targeting sequence of human cytochrome c oxidase (COX4)[33,34]. Following pulse-SILAC and proximity labeling, biotinylation was visualized with immunofluorescence, which showed highly colocalized signal between biotinylated proteins and the mitochondrial marker TOMM20 (Fig. 1B). To quantify protein turnover, we chose 4 h, 8 h and 12 h as the duration of pulse-SILAC labeling and performed

replicated experiments for each time point. Following proximity labeling, cells were lysed and biotinylated proteins were enriched with streptavidin-coated beads. Both western blot analysis and silver staining confirmed successful biotinylation and protein enrichment across replicates at 8 h and 12 h (Supplementary Fig. 1). For the 4 h experiment, silver staining revealed low protein enrichment efficiency, but sufficient amount of peptides were obtained for subsequent MS analysis (Supplementary Fig. 1). In negative control samples omitting APEX2, the biotin-phenol substrate, or hydrogen peroxide, only endogenous biotinylated proteins were detected (Supplementary Fig. 2). Enriched proteins were digested with trypsin and analyzed by LC-MS/MS.

A total of 229, 612 and 410 proteins were identified and quantified across 4-, 8- and 12-h replicated pulse-SILAC experiments, respectively. The overlap of these three groups yielded a list of 183 proteins (Supplementary Fig. 3, Supplementary Data 1), including 162 (85%) proteins annotated in the Uniprot database, which has an established inventory of mitochondrial proteins (Fig. 1C, Supplementary Fig.4). This level of spatial specificity is comparable to previous reports[26]. Notably, the coverage of mitochondrial proteome is lower than the previous report using mito-APEX[26], which we attribute to differences in the sample preparation workflow and LC-MS/MS methods between two experiments (see Methods). As a further demonstration of mitochondrial matrix specificity, proteins identified by mito-APEX2 are mapped to the structure of respiratory complexes in the inner mitochondrial membrane (IMM), revealing that only matrix-exposed protein subunits are captured by mito-APEX2 labeling (Fig. 1G). This observation is consistent with the view that IMM restricts the diffusion of phenoxyl free radicals. For each quantified protein, we define its metabolic replacement fraction as $MRF = H/L\ /\ (1 + H/L)$, where $H/L$ is the measured SILAC ratio. The calculated MRF values are highly correlated between replicates, with Pearson's correlation coefficients ranging between 0.90 and 0.95 (Fig. 1D and Supplementary Fig. 5). Taken together, the above analysis demonstrates the high spatial specificity, good proteomic coverage, and high reproducibility of prox-SILAC method.

As we increased the duration of pulse-SILAC, the overall MRF values increased substantially, from an average of $22 \pm 11\%$ (mean ± s.d.) at 4 h to $40 \pm 10\%$ at 12 h (Fig. 1D and Supplementary Fig. 5). At each time point, we observed a broad and asymmetric distribution of MRFs: whereas a majority of proteins have low MRFs, a small subset of proteins exhibits distinctively higher MRFs (Fig.1E). For example, the highest and lowest MRFs measured at 4 h ($MRF_{4h}$) differ by as much as 7-fold ($79 \pm 2\%$ vs. $11 \pm 4\%$) (Fig. 1E, Supplementary Data 1). Since, at steady state, the abundance of cellular proteins should double per cell cycle, the baseline MRF is expected to be approximately 2% per hour. Indeed, the slowest MRFs, as observed in several metabolic enzymes (e.g. FADH1 and ETFB), occur at the low level of 3% per hour. Intrigued by the broad MRF distribution, we divided our dataset into three categories: "high MRF" (>25%), "medium MRF" (between 10% and 25%), and "low MRF" (<10%), according to their MRF values measured at 8 h ($MRF_{8h}$). Gene Ontology analysis reveals that proteins involved in the electron transport chain tend to have higher MRFs, whereas proteins related to mitochondrial organization and lipid metabolism are characterized with low MRFs (Supplementary Fig. 6).

To further analyze the heterogeneity of MRF values within protein complexes, we mapped $MRF_{8h}$ to the structures of respiratory complexes (RC) I to V (Fig. 1G, Supplementary Data 1). Protein components in RC-I overall have the highest $MRF_{8h}$ ($57 \pm 16\%$), whereas those in RC-V have the lowest ($35 \pm 14\%$) (Supplementary Fig. 7, Supplementary Data 1). This observation agrees with the conclusion drawn from a previous study using HeLa cells as a model[17]. Within RC-I, we observed the highest $MRF_{8h}$ for protein subunits NDUFA13, NDUFB8 and NDUFB9, with an average of $78 \pm 6\%$. Similarly, we observed large variations in $MRF_{8h}$ in protein components of the mitochondrial

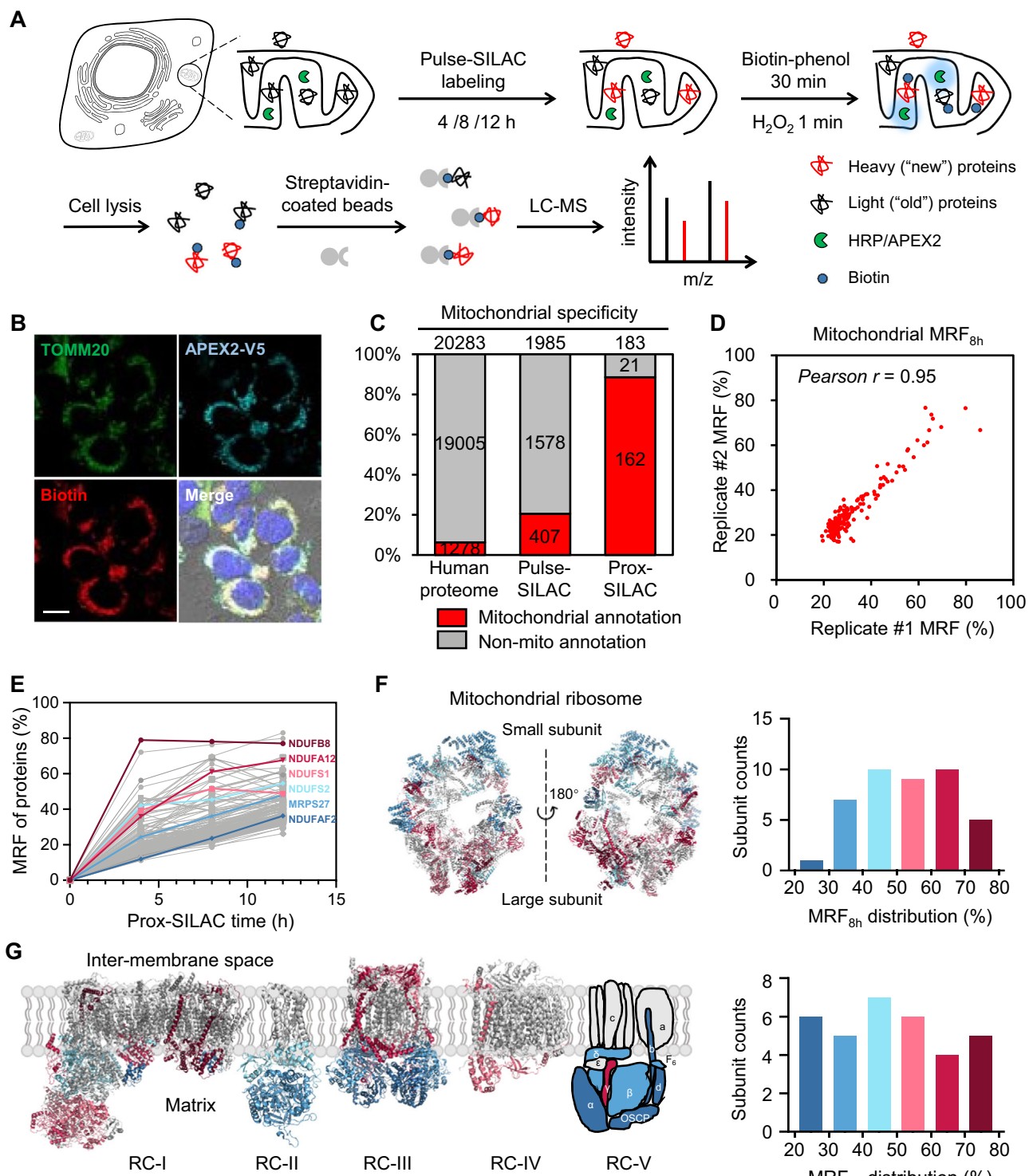

**Fig. 1 | Characterization of prox-SILAC method and its application to mitochondria. A** Experimental scheme of prox-SILAC with APEX2 targeted to the mitochondrial matrix (mito-APEX2). Newly synthesized proteins in the time windows of 4 to 12 h are tagged with heavy isotope-encoded lysine and arginine. Following APEX2 labeling, biotinylated proteins are enriched via affinity purification, digested with trypsin, and quantified via LC-MS/MS analysis. **B** Representative confocal immunofluorescence images of HEK293T cells showing the localizations of APEX2 (anti-V5), mitochondria (anti-TOMM20), and biotinylated proteins (streptavidin-AlexaFluor 647). Scale bar = 10 μm. Images were acquired from at least three fields of view in three biological replicates. **C** Specificity analysis for prox-SILAC mitochondrial proteome. Red bars and numbers indicate proteins with prior mitochondrial annotations in the GOCC database. Left: human proteome (from Uniprot database); middle: proteins identified in replicated pulse-SILAC experiments; right: proteins identified in replicated prox-SILAC experiments. **D** Scatter plots showing the $MRF_{8h}$ values of mitochondrial proteins identified in replicated prox-SILAC experiments at three time points. **E** Plot of MRF traces with protein examples highlighted in colors. **F**, **G** Mapping of $MRF_{8h}$ to the structure of the mitochondrial ribosome (PDB ID: 3J9M) (**F**) and the respiratory complexes I (5XTD), II (6VAX), III (5XTE), IV (5Z62), and V (cartoon representation[26]) (**G**). The histogram distribution and color map of $MRF_{8h}$ values are shown on the right.

ribosome (Fig. 1F, Supplementary Data 1), with proteins in the large subunit having higher $MRF_{8h}$ ($58 \pm 14\%$) than proteins in the small subunit ($47 \pm 11\%$). Mapping the $MRF_{8h}$ values to the mitochondrial ribosome structure reveals patches of high and low MRF regions, indicating that neighboring proteins tend to share similar turnover dynamics (Fig. 1F).

We also performed pulse-SILAC experiments with mito-APEX2 cell line to evaluate the MRF values at the whole-cell level. From three replicated 8-h pulse-SILAC experiments, we identified and quantified 1584 proteins from at least two replicates (Supplementary Fig. 8a, 9, Supplementary Data 1). For the 111 proteins identified in both prox-SILAC and pulse-SILAC experiments (Supplementary Fig. 8b), their overall $MRF_{8h}$ values are similar between the two datasets (Supplementary Fig. 10). The notable exceptions are several proteins involved in the oxidative phosphorylation pathway (i.e. OXPHOS subunits): NDUFB9 ($76 \pm 7\%$ in prox-SILAC vs. $27 \pm 7\%$ in pulse-SILAC), ATP5C1 ($69 \pm 1\%$ vs. $32 \pm 1\%$) and COX5A ($56 \pm 2\%$ vs. $34 \pm 2\%$) (Supplementary Figure 10). The pulse-SILAC $MRF_{8h}$ values of OXPHOS components range from 26% to 56% (mean $MRF_{8h}$ value 34%), which are lower than the corresponding prox-SILAC $MRF_{8h}$ values (from 24% to 78%, mean value 50%).

## Profiling ER protein turnover with prox-SILAC

Having established our prox-SILAC method in the mitochondrial matrix, we next focused on the ER, which is a hub of secretory pathway proteins (Supplementary Fig. 11). To label ER proteins, we chose HRP instead of APEX2 for its higher labeling efficiency in the oxidizing environment of ER lumen[23]. HRP is targeted to the ER lumen via both *N*-terminal fusion with the signal sequence derived from immunoglobin Igκ and *C*-terminal fusion with the ER retention motif KDEL (ss-HRP-KDEL). Immunofluorescence imaging of HEK293T cells stably expressing ss-HRP-KDEL confirmed the good co-localization between protein biotinylation and the ER marker, calnexin (Fig. 2A).

We performed duplicated prox-SILAC experiments with isotope tagging for 4 h, 8 h and 12 h. Both western blot and silver staining confirmed successful protein biotinylation and affinity enrichment (Supplementary Fig. 12). LC-MS/MS analysis identified a total of 510, 210 and 265 proteins across replicated experiments of 4-, 8- and 12-h prox-SILAC, respectively, with an overlap of 186 proteins (Fig. 2B and Supplementary Fig. 13, Supplementary Data 2). Gene Ontology cellular component (GOCC) analysis revealed a remarkable secretory pathway specificity of 97% (Fig. 2B). Similar to the case of mitochondrial matrix,

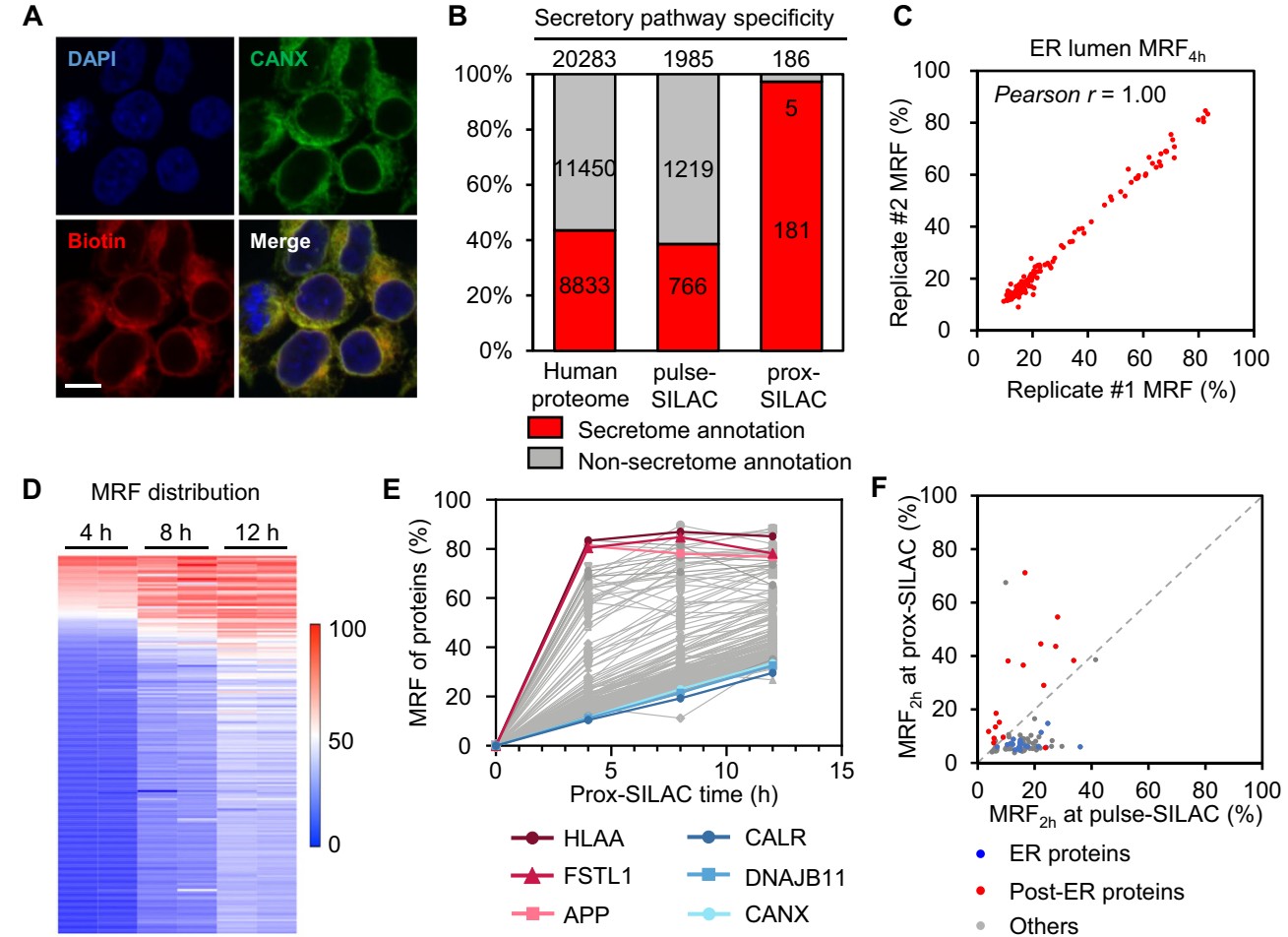

**Fig. 2 | Mapping ER protein turnover with prox-SILAC. A** Representative confocal immunofluorescence images showing the localizations of biotinylated proteins (streptavidin-AlexaFluor 647), ER marker (anti-Calnexin), and the nucleus (DAPI) in HEK293T cells expressing ss-HRP-KDEL. Scale bar = 10 μm. Images were acquired from at least three fields of view in three biological replicates. **B** Specificity analysis of the ER proteome. Proteins with prior secretory pathway annotations in the GOCC database are indicated in red[28]. Left: entire human proteome (from Uniprot database); middle: proteins identified in replicated pulse-SILAC experiments; right: proteins identified in replicated prox-SILAC experiments. **C** Scatter plot showing $MRF_{4h}$ values of labeled proteins in replicated experiments at three time points. **D** Heatmap depicting the MRF values of identified ER proteins in replicated experiments at three time points. **E** Plot of MRF traces with protein examples highlighted in colors. **F** Scatter plot of $MRF_{2h}$ values of ER proteins measured in ER prox-SILAC against whole-cell pulse-SILAC. The ER proteins are identified in all prox-SILAC replicates at five time points and the 2-hr whole-cell pulse-SILAC replicates. Red dots indicate post-ER trafficking proteins (i.e. targeted to the cell membrane, Golgi apparatus or lysosome), while blue dots indicate ER resident proteins.

the MRF values for identified ER proteins are highly correlated between replicated experiments at the three time points, with Pearson's correlation coefficients ranging between 0.96 and 1.00 (Fig. 2C and Supplementary Fig. 14). Interestingly, the $MRF_{8h}$ values for ER proteins ($38 \pm 18\%$) are overall substantially higher than mitochondrial matrix proteins ($31 \pm 12\%$) (Fig. 2D, E). This phenomenon may be attributed in part to the anterograde protein trafficking from the ER to the Golgi apparatus, thus causing a rapid clearance of "old" secretory pathway proteins in the ER lumen.

According to this model, ER resident proteins such as chaperones would have significantly lower MRFs than trafficking proteins such as plasma membrane receptors (Fig. 2E). To test this model, we repeated the above ER prox-SILAC experiments by focusing on shorter time windows (i.e. 1 h and 2 h) of isotope labeling, to more accurately characterize the MRF values of ER proteins (Supplementary Fig. 14). Indeed, the $MRF_{2h}$ values of post-ER secretory pathway proteins ($27 \pm 20\%$) are on average 2-fold higher than those of ER resident proteins ($14 \pm 10\%$). In addition, we calculated the half-lives ($t_{1/2}$) of the 183 proteins identified across all five time points. Their $t_{1/2}$ values are broadly distributed from 0.9 hr to 24.7 h, with a mean $t_{1/2}$ of 14.3 h (Supplementary Data 2, Supplementary Fig. 15). The protein with the shortest $t_{1/2}$ (0.9 h) in the ER is amyloid-beta precursor protein (APP), a cell surface receptor relevant to neurite growth and neuronal adhesion. On the contrary, the protein with the longest $t_{1/2}$ (24.7 h) is calreticulin, an ER resident protein.

Notably, the measured MRF values reflect the rates of both protein degradation and trafficking. To further evaluate the contribution from protein degradation, we analyzed the turnover dynamics of whole-cell proteome. In two replicated pulse-SILAC experiments, we identified and quantified the $MRF_{2h}$ values for 2253 proteins in the whole cell lysate (Supplementary Fig. 16), among which, 89 proteins overlapped with our ER prox-SILAC dataset (Supplementary Fig. 17). When comparing their $MRF_{2h}$ values measured in either whole-cell pulse-SILAC or ER prox-SILAC, we observed distinct patterns between post-ER trafficking proteins (e.g. proteins destined for the plasma membrane, Golgi apparatus or lysosome) and ER resident proteins (Fig. 2F). The $MRF_{2h}$ values of post-ER trafficking proteins are significantly higher in prox-SILAC than in pulse-SILAC, indicating that their rapid clearance from the ER is caused by trafficking rather than by degradation. Since many cellular proteins are constantly trafficking between multiple subcellular compartments, it would be more informative to measure protein turnover at the subcellular level, which may differ substantially from those measured at the whole-cell level.

To further investigate protein turnover on the plasma membrane, we performed 2-hr pulse-SILAC labeling in HEK293T cells and labeled the cell surface protein population with the membrane-impermeant chemical reagent, Sulfo-NHS-LC-Biotin. Similar to our workflow of prox-SILAC, biotinylated proteins were subsequently enriched with streptavidin-coated beads and analyzed by LC-MS/MS to derive the $MRF_{2h}$ values (Supplementary Fig. 18). In two replicated experiments, we identified and quantified 508 cell surface proteins (Supplementary Fig. 19). For the 34 proteins identified in both ER lumen and cell surface prox-SILAC datasets, their $MRF_{2h}$ values differ substantially (Supplementary Fig. 20). Membrane receptors and ion pumps such as TFRC ($71 \pm 1\%$ vs. $4 \pm 0\%$), ATP1A1 ($68 \pm 1\%$ vs. $19 \pm 0\%$) and IGF2R ($55 \pm 3\%$ vs. $5 \pm 3\%$), exhibit much higher ER lumen $MRF_{2h}$ values than cell surface $MRF_{2h}$, which is attributed to the time required for them to traffic from the ER lumen to the plasma membrane. To our surprise, two protein chaperones, SERPINH1 ($5.0 \pm 0\%$ vs. $86 \pm 0\%$) and PDIA6 ($4 \pm 5\%$ vs. $56 \pm 16\%$) exhibit substantially lower ER $MRF_{2h}$ values than cell surface $MRF_{2h}$. SERPINH1 (Hsp47) is a collagen-specific molecular chaperone and plays an important role in the collagen biosynthesis[35]. While PDIA6 mainly functions as a chaperone that inhibits the aggregation of misfolded proteins during unfolded protein response (UPR)[36], it has also been reported to bind integrin β3 subunit on the cell surface to promote platelet activation[37]. We thus speculate that the higher MRF values of protein chaperones may be associated with their distinct functions on the cell surface. Together, the above comparison of prox-SILAC at various subcellular compartments highlight its advantage in investigating spatially resolved protein turnover.

## Mapping subcellular protein turnover under ER stress

ER stress response, which can be induced through multiple pathways including defective protein folding, has been implicated in the onset and the progression of an array of diseases ranging from cancer to neurodegeneration[38]. It has been known that proteome homeostasis changes considerably when cells are under stress. Specifically, the rates of protein synthesis, degradation and trafficking would be tuned to better cope with the stress condition. To depict a detailed picture of protein turnover remodeling, we applied prox-SILAC to HeLa cells under ER stress induced by thapsigargin, a small molecule inhibitor of $Ca^{2+}$ transport in the ER (Fig. 3A, B). Successful induction of ER stress is confirmed by the up-regulation of the chaperone marker BiP (Supplementary Fig. 21).

We performed duplicated ER prox-SILAC experiments for both thapsigargin-treated (2 h) and control samples, identifying 306 and 320 proteins, respectively (Fig. 3C). The overlapping 265 proteins have a secretory pathway specificity of 97% (Fig. 3D and Supplementary Fig. 22, Supplementary Data 3). Compared to the control sample, ER stress causes a global down-regulation of protein turnover in the ER (Fig. 3E), which is consistent with the findings of protein translational shutdown in the previous study of UPR[39,40] and ER-associated degradation (ERAD)[41] pathways. Notably, against this slow turnover background, we observed significantly elevated $MRF_{2h}$ values for a handful of proteins, including HSPA5, HSPA6, JAGN1 and LMAN1 (Fig. 3E). For example, the $MRF_{2h}$ of protein chaperones HSPA5 and HSPA6 increased from approximately $4 \pm 0\%$ at the basal level to $15 \pm 1\%$ and $16 \pm 0\%$ under ER stress, respectively. It has been reported that the activation of UPR pathways promotes the expression of chaperones including HSP proteins and BiP[42]. Consistent with this view, our observed increase in $MRF_{2h}$ indicates the rapid entry of these proteins into the ER lumen. Similarly, the observed $MRF_{2h}$ for mannose-specific lectin LMAN1 doubled from $5 \pm 0\%$ to $10 \pm 1\%$ upon ER stress, which is in agreement with stress-induced LMAN1 transcription[43] and its enrichment in the Golgi[44], thus causing rapid replacement of the ER population with newly synthesized LMAN1 protein.

We also performed 2-h pulse-SILAC experiments using HeLa-SS-HRP-KDEL cell line with or without thapsigargin treatment. We quantified the $MRF_{2h}$ values of 1429 and 1598 proteins in three replicates of thapsigargin-treated and control samples, respectively, resulting in 1202 overlapped proteins (Supplementary Fig. 23-24, Supplementary Data 3). We extracted the secretory pathway proteins from the pulse-SILAC experiments using GOCC annotations, generating a pulse-SILAC x subcellular information dataset of 584 proteins. Overall, much higher $MRF_{2h}$ values were observed for post-ER trafficking proteins in the prox-SILAC experiments than pulse-SILAC (Supplementary Fig. 25). While ER stress causes a significant global reduction of $MRF_{2h}$ values measured in prox-SILAC (mean values 18% in thapsigargin-treated vs. 26% in the control), the decrease is less noticeable in pulse-SILAC (13% in thapsigargin-treated vs. 14% in the control) (Fig. 3E).

## Mapping subcellular protein turnover during neurite growth

The cellular proteomic landscape changes dramatically during differentiation[8]. As a neuroblastoma-derived cell line, SH-SY5Y cells could be induced to differentiate into mature neuron-like cells and are often used as a Parkinson's disease model[45,46]. Since neurite growth is

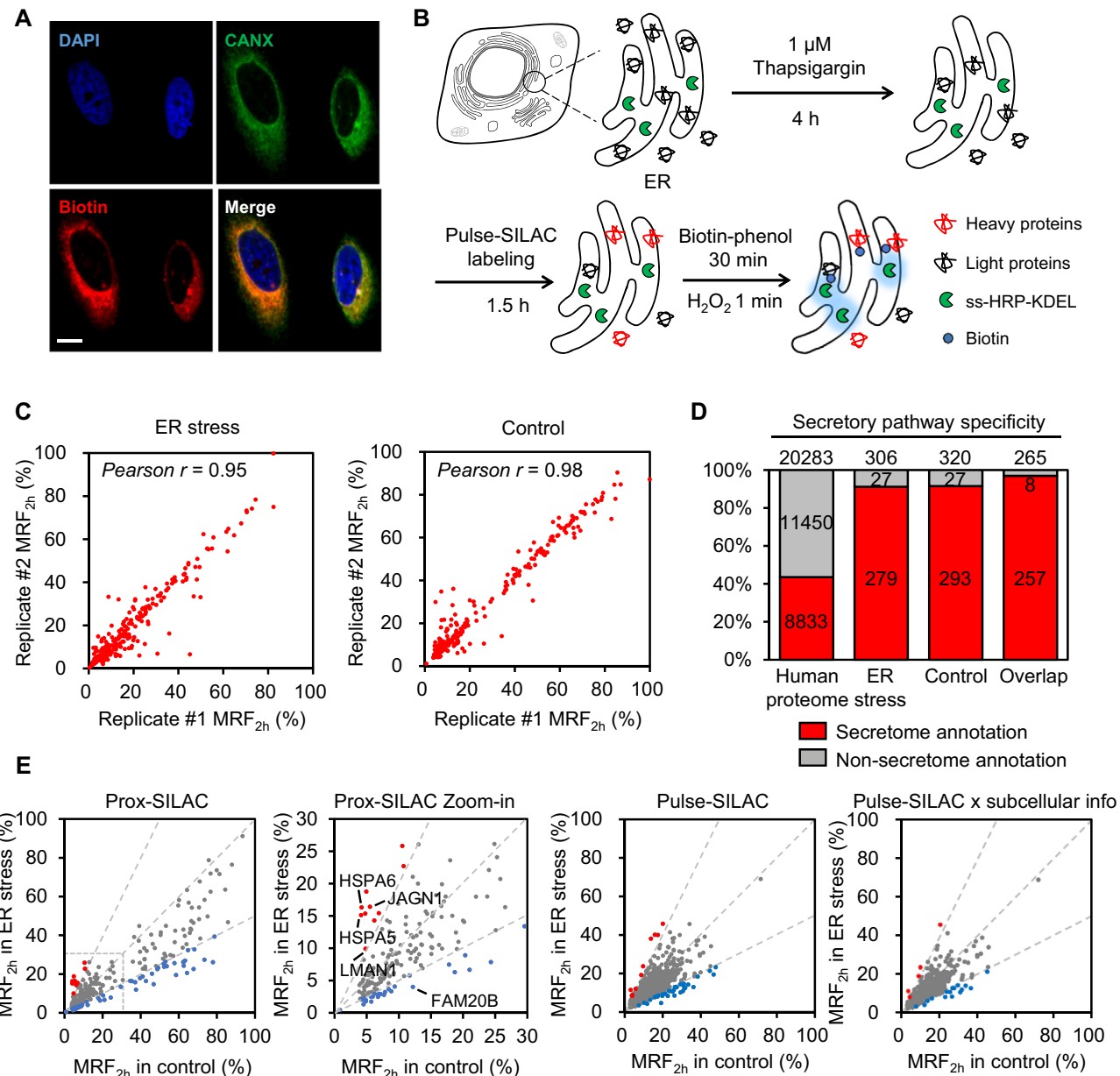

**Fig. 3 | Analysis of protein turnover changes under ER stress. A** Representative confocal immunofluorescence images showing the localizations of biotinylated proteins (streptavidin-AlexaFluor647), ER marker (anti-Calnexin), and the nucleus (DAPI) in HeLa cells expressing ss-HRP-KDEL proteins. Scale bar = 10 μm. Images were acquired from at least three fields of view. **B** Experimental scheme of ER stress induction and ER prox-SILAC labeling in HeLa cells. Cells are treated with 1 μM thapsigargin for 4 h prior to prox-SILAC. **C** Scatter plots showing MRF$_{2h}$ values of labeled proteins in ER stress (left) and the control (right). **D** Specificity analysis of proteomic data in ER stress and the control experiments. Proteins with prior secretory pathway annotations in the GOCC database are indicated in red. **E** Scatter plot comparing the MRF$_{2h}$ values in ER stress versus the control sample. Red and blue dots represent proteins with 2-fold higher or lower MRF$_{2h}$ values, respectively. From left to right: proteins in prox-SILAC experiments, a zoom-in view of prox-SILAC data, proteins in pulse-SILAC experiments, proteins with secretome annotations in pulse-SILAC experiments.

often accompanied by a rapid turnover of secretory pathway proteins involved in membrane structural maintenance, we sought to apply ER prox-SILAC to quantify the protein turnover changes in SH-SY5Y cells undergoing stimulated neurite growth. We created an SH-SY5Y cell line stably expressing ss-HRP-KDEL. Confocal immunofluorescence imaging confirmed the co-localization between HRP-mediated protein biotinylation and the ER marker calnexin (Supplementary Fig. 26). To induce neurite growth, we treated SH-SY5Y cells with 10 μM all-trans retinoic acid (ATRA)[47–49] for 7 days, followed by applying brain-derived neurotrophic factor (BDNF)[50] for another 7 days (Fig. 4A). At the end of the 14-day differentiation protocol, the neurite length increased from 20 ± 6 μm to 53 ± 12 μm (Fig. 4A).

We focused on the early stage of differentiation and performed duplicated prox-SILAC experiments at days 0 (D0), 7 (D7) and 10 (D10), with cells pulse-labeled with heavy isotope-encoded lysine and arginine for 2 h. A total of 486, 356 and 314 proteins were identified and quantified at D0, D7 and D10, respectively (Fig. 4B, Supplementary Data 4), with secretory pathway specificity ranging between 90% and 94% (Supplementary Fig. 27). Overall, as cells undergo differentiation, protein turnover in the ER lumen gradually decreased, as revealed by the leftward shift in the cumulative distribution curve of their MRF$_{2h}$ values (Fig. 4C).

For the 291 proteins identified in both D0 and D7 experiments, 39 proteins turnover at least 2-fold faster at D7 than at D0, while 17

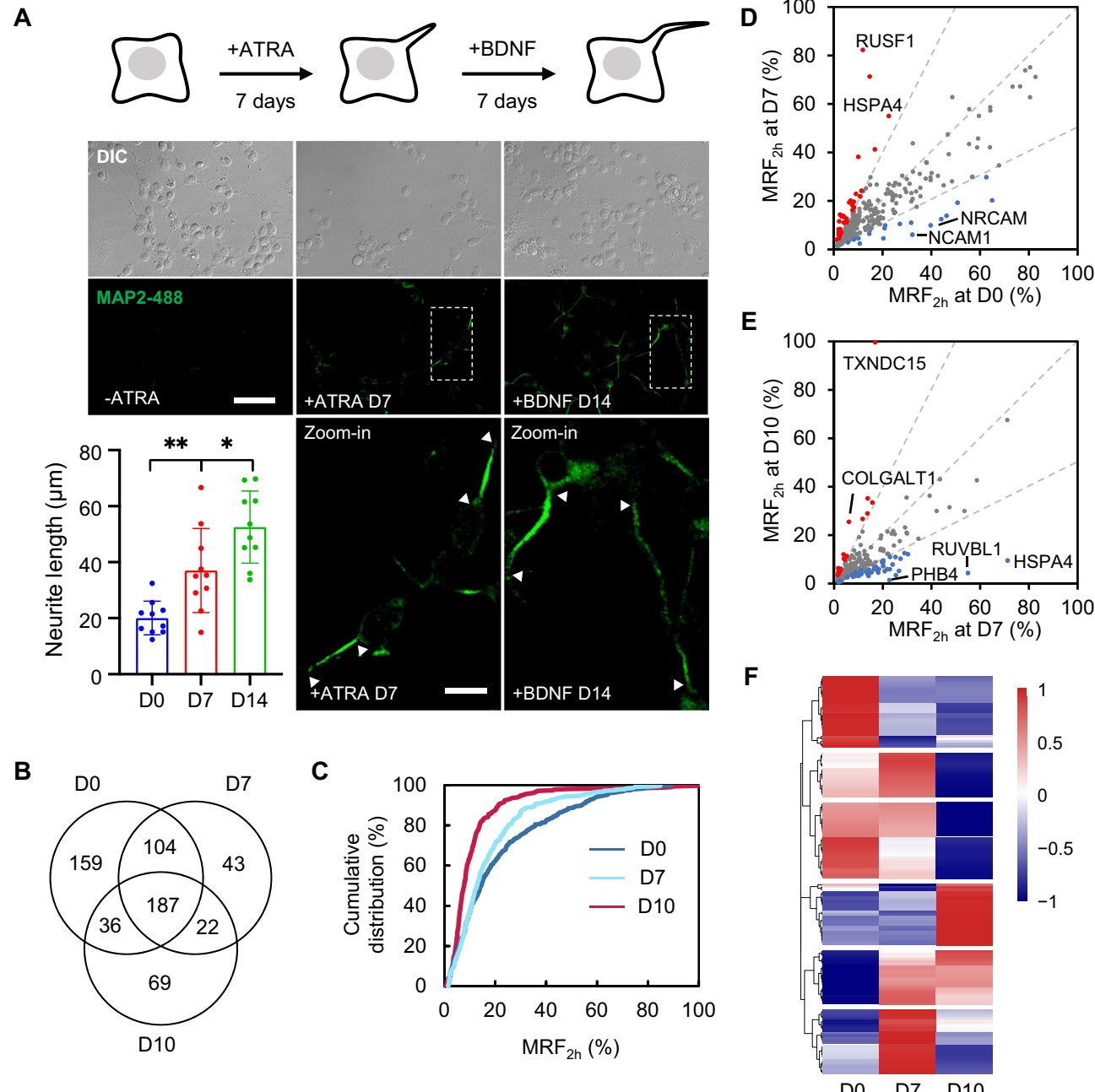

**Fig. 4 | Profiling the turnover changes of ER proteins during neurite growth.** **A** Schematic time course (top) and immunofluorescence images (bottom) of SH-SY5Y differentiation. Neurites are visualized via staining with anti-MAP2 antibody. The proximal and distal ends of neurites are indicated as white triangles. Scale bar = 50 μm. Statistics are shown at the bottom left, with $n = 10$ cells per sample and $p$ values calculated from two-sided Student's $t$-test. *$p$ value between D7/D10 is 0.0238; **$p$ value between D0/D7 is 0.0038. Source data are provided as a Source Data file. **B** Venn diagrams showing the numbers of quantified proteins at D0, D7 and D10 replicated experiments. **C** Cumulative distribution curves of $MRF_{2h}$ values at D0, D7 and D10. **D, E** Scatter plots showing changes in $MRF_{2h}$ values of ER proteins between D0 and D7 (**D**) and between D7 and D10 (**E**). Red and blue dots represent proteins with 2-fold higher and lower $MRF_{2h}$, respectively. **F** Heatmap showing normalized z-scores of $MRF_{2h}$ values along differentiation time line. ER proteins are divided into six categories according to their patterns of $MRF_{2h}$ changes.

proteins turnover at least 2-fold slower (Fig. 4D, Supplementary Data 4). Notably, the differentiation protocol has dramatically stimulated the turnover of the secreted protein neudesin, which is a neuron-derived neurotrophic factor ($MRF_{2h}$ from $2 \pm 0\%$ at D0 to $12 \pm 3\%$ at D7). We also observed increased $MRF_{2h}$ for several protein post-translational modification enzymes, including prenylcysteine oxidase 1 ($2 \pm 0\%$ to $8 \pm 0\%$), prolyl 3-hydroxylase 1 ($3 \pm 0\%$ to $14 \pm 4\%$), peroxiredoxin-4 ($2 \pm 0\%$ to $7 \pm 1\%$), and peptidyl-prolyl cis-trans isomerase B ($2 \pm 0\%$ to $7 \pm 1\%$). Interestingly, the highest increase in

$MRF_{2h}$ was observed for RUS family member 1 (RUSF1, $12 \pm 2\%$ to $82 \pm 11\%$), previously known as C16orf58, whose function in neuronal differentiation has remained elusive. Among the "slowly turnover" group are several members of adhesion proteins, including neural cell adhesion molecule L1 (N-CAM-L1, from $51 \pm 2\%$ at D0 to $19 \pm 1\%$ at D7), neural cadherin (N-cadherin, $46 \pm 6\%$ to $14 \pm 3\%$), neuronal cell adhesion molecule (Nr-CAM, $40 \pm 3\%$ to $10 \pm 0\%$), and neural cell adhesion molecule 1 (N-CAM-1, $32 \pm 2\%$ to $6 \pm 3\%$). The above data suggest that ATRA-induced first-stage differentiation has set a brake

to adhesion molecules, presumably as a means of coordinating cellular motility with morphological changes.

During the second stage of differentiation (from D7 to D10), 209 proteins are identified and quantified across a total of four replicated experiments (Supplementary Data 4). Within only three days, the $MRF_{2h}$ values of 73 proteins change by at least 2-fold, with 19 proteins turning over faster and 54 protein slower (Fig. 4E). Against this general trend of slower turnover, the cilium-targeted thioredoxin domain-containing protein 15 (TXNDC15) exhibits the highest increase in $MRF_{2h}$, from a modest ratio of $17 \pm 0\%$ at D7 to almost complete turnover ($100 \pm 0\%$) at D10. Notably, the $MRF_{2h}$ of several enzymes involved in collagen biosynthesis significantly increase from D7 to D10, including collagen-binding SERPINH1 ($2 \pm 0\%$ to $6 \pm 0\%$), multifunctional procollagen lysine hydroxylase and glycosyltransferase LH3 (PLOD3, $4 \pm 0\%$ to $10 \pm 1\%$), and prolyl 4-hydroxylase subunits (P4HA1, $4 \pm 1\%$ to $10 \pm 0\%$; P4HA2, $10 \pm 1\%$ to $19 \pm 4\%$). We also observed increased turnover for proteins involved in calcium-dependent activities, including calreticulin ($2 \pm 0\%$ to $4 \pm 1\%$) and reticulocalbins (RCN1, $2 \pm 0\%$ to $4 \pm 1\%$; RCN2, $2 \pm 0\%$ to $4 \pm 1\%$; RCN3, $5 \pm 1\%$ to $9 \pm 0\%$).

Finally, for the overlap of 187 proteins across six replicated experiments performed at D0, D7 and D10, their $MRF_{2h}$ values are clustered into 6 categories (Fig. 4F, Supplementary Data 4). Category I, which includes several cell adhesion molecules (e.g. N-CAM-L1, N-CAM-1, Nr-CAM), features steadily decreasing $MRF_{2h}$ values throughout the differentiation process. In contrast, the $MRF_{2h}$ values for category IV proteins gradually increase during differentiation. Members in this category include prolyl 4-hydroxylase subunits (P4HA1, P4HA2 and P4HB), peptidyl-prolyl cis-trans isomerases (FKBP7 an FKBP10), and reticulocalbins (RCN1, RCN2 and RCN3). Interestingly, category VI proteins feature a peak in their MRFs at D7, which account for 14% (26 out of 187) proteins in our prox-SILAC dataset. Together, the above data demonstrate the power of prox-SILAC to resolve protein turnover dynamics in the model of neuronal differentiation. Our data shed light on the cellular capacity of remodeling protein turnover dynamics as a means of fine-tuning the membrane proteome, which may serve to better adapt to morphological changes such as neurite growth.

In parallel, we performed 2-h pulse-SILAC experiments at D0, D7 and D10 during neurite growth, resulting in the identification of 1687, 1381 and 1496 proteins at these time points, respectively (Supplementary Fig. 28-29, Supplementary Data 4). Similar to our previous comparisons between prox-SILAC and pulse-SILAC in the ER, overall higher $MRF_{2h}$ values are observed in prox-SILAC at D0 and D7, particularly for post-ER trafficking proteins (Supplementary Fig. 30). However, the difference is much smaller at D10, where the $MRF_{2h}$ values are lower in both prox-SILAC and pulse-SILAC (Supplementary Fig. 30).

During the early stage of differentiation (from D0 to D7), the $MRF_{2h}$ of neudesin increases by 5-fold in prox-SILAC experiments (from $2 \pm 0\%$ to $12 \pm 3\%$) (Fig. 4D), yet its pulse-SILAC $MRF_{2h}$ suffers nearly 2-fold decrease, from $29 \pm 16\%$ to $16 \pm 7\%$ (Supplementary Fig. 31). Prolyl 3-hydroxylase 1 and N-CAM-1 also exhibit opposite $MRF_{2h}$ changes between prox-SILAC and pulse-SILAC (Supplementary Data 4). In the second stage of differentiation from D7 to D10, while the $MRF_{2h}$ values of most proteins become lower (downward deviation from the diagonal in Fig. 4E), the trend is less noticeable in the corresponding pulse-SILAC dataset or the pulse-SILAC x subcellular information dataset (Supplementary Fig. 31).

## Discussion

To summarize, we have developed prox-SILAC method that combines peroxidase-mediated proximity labeling with pulse-SILAC tagging of newly synthesized proteins. Conventional pulse-

SILAC labeling on the whole cell level reveals the protein turnover dynamics, which includes protein synthesis and degradation. In contrast, the MRF values measured with prox-SILAC reflects the combine effects of not only protein synthesis and degradation, but also local trafficking at specific subcellular localizations. In the past, subcellular protein turnover dynamics was measured by combining fractionation with pulse SILAC labeling. This has been successfully implemented for organelles that can be readily purified, such as the mitochondria[17] and the nucleus[19]. However, such approach could not be easily extended to other compartments, including the highly tubular ER structure. Isolated microsome was used to profile ER proteome, but generating incomplete datasets with poor overlap in independent works[51,52]. Over the past decade, APEX2 and HRP have been applied to resolve the proteome in multiple subcellular compartments, including both membrane-bound space (e.g. ER-PM junction[53], synaptic cleft[54], etc.) and membraneless condensates (e.g. stress granule[55], nuclear speckle[56], etc.). We envision that prox-SILAC could add to this toolbox by providing protein turnover information, in addition to protein abundance.

In the mitochondrial matrix, we have demonstrated the high spatial specificity of proximity labeling and high accuracy of protein turnover quantitation. In the mitochondria of HEK293T cells, we observed heterogeneous distribution of protein turnover dynamics in respiratory complex I-V subunits, with an overall higher MRF in RC-I and lower in RC-V. The observed differences in $MRF_{8h}$ values of OXPHOS subunits between pulse-SILAC (average 34%) and prox-SILAC (average 50%) may be attributed to protein translocation. Our data suggest that a small sub-population of OXPHOS proteins, such as NDUFB9 and ATP5C1, may not be accessible to mitochondrial matrix-targeted APEX2 (e.g. on the outer membrane as newly synthesized protein, or outside mitochondria), although we do not have experimental data to support this view.

Notably, remarkable variations in the MRF values of protein subunits within mitochondrial ribosome complexes and respiratory complexes have also been reported in previous work via pulse-SILAC labeling in HeLa cells[17]. In these studies, the H/L ratios exhibited a 5-fold difference in ribosomal proteins and a 4-fold difference in respiratory complex proteins, indicating that these proteins are synthesized and imported into mitochondria in considerable excess over the minimum amount required to support assembly. The above observation reveals a complicated assembly process that warrants further investigation. As OXPHOS components are liable to be oxidized by leaked electrons from the electron transport chain, excessive copies in the mitochondria may act as a reserve to quickly replace the damaged subunits, thus keeping the OXPHOS machineries functioning smoothly.

As the MRF values of duplicated prox-SILAC experiments are highly correlated with Pearson's correlation coefficients >0.9, the mean MRF values are used for subsequent analysis. Notably, negative changes in MRF values during the prox-SILAC time course were observed for some of the proteins, which typically happens when their MRF values are high or almost at saturated levels. Similar trend has also been reported in previous pulse-SILAC studies[17], which could be attributed to measurement errors of mass spectrometry quantitation. To avoid measurement near the MRF saturation level, we focused on an early time point of 2 h for the subsequent prox- and pulse-SILAC experiments.

Application of prox-SILAC to the ER lumen reveals distinct distribution of MRF for post-ER trafficking proteins versus ER resident proteins. The substantially higher MRF values of post-ER trafficking proteins are consistent with the model of protein trafficking in the secretory pathway. Such distinction could not be achieved with whole-cell pulse-SILAC methods, thus highlighting the unique

advantage of the subcellular spatial resolution provided by prox-SILAC.

When cells are challenged with ER stress, UPR and ERAD pathways are activated to alleviate stress, leading to global translational shutdown and the expression of specific proteins involved in stress-response pathway[57]. Indeed, our prox-SILAC dataset reveals a general trend of suppressed protein turnover during thapsigargin-induced ER stress. Exceptions to this trend include several proteins relevant to ER stress response (e.g. heat shock protein chaperones), which exhibit substantially elevated MRFs under stress condition as compared to physiological condition. For example, the ER resident protein involved in vesicular trafficking, JAGN1 (protein jagunal homolog 1), has recently been implicated in ER stress response[58]. In our dataset, the $MRF_{2h}$ of JAGN1 increases by nearly 3-fold (from $6 \pm 0\%$ to $16 \pm 3\%$) upon ER stress.

Finally, we applied prox-SILAC to investigate protein turnover dynamics in cells undergoing stimulated differentiation, using neuroblastoma cell line SH-SY5Y as a model. SH-SY5Y cells have been commonly used for studying neural differentiation and the pathogenesis of Parkinson's disease[59]. Our data reveal a gradual decrease in protein turnover as cells progress through ATRA/BDNF-induced neurite growth. This trend is particularly evident for cell adhesion proteins, which are involved in regulating cell motility. In contrast, several secretory pathway enzymes involved in protein maturation (i.e. post-translational modifications, prolyl cis-trans isomerization, etc.) tend to slowly increase their turnover during differentiation. Together, the above examples demonstrate the power of prox-SILAC for mapping subcellular protein turnover dynamics in both acute (on the time scale of hours) stress response and long-term (on the time scale of days) adaptation scenarios.

For all the above prox-SILAC experiments, we have performed the equivalent pulse-SILAC experiments at the whole-cell level. Prox-SILAC offers unique coverage of the subcellular proteome as compared to pulse-SILAC. For example, our mitochondrial matrix prox-SILAC dataset contains 72 proteins that are missed by pulse-SILAC, despite the overall larger coverage of the latter (407 vs. 162 mito proteins). Similarly, our ER-lumen prox-SILAC datasets in HEK293T cells, HeLa cells, and thapsigargin-stressed HeLa cells have uniquely identified 94, 186, and 172 proteins, respectively, as compared to the pulse-SILAC counterparts. In SH-SY5Y cells undergoing differentiation, a total of 262, 180 and 90 proteins are uniquely quantified in ER lumen prox-SILAC at D0, D7 and D10, respectively, as compared to the corresponding pulse-SILAC datasets.

Prox-SILAC also provides the high spatial specificity required for resolving subcellular MRF values. A comparison between our ER lumen prox-SILAC versus ER lumen pulse-SILAC datasets reveals that post-ER trafficking proteins overall have substantially higher MRF values in prox-SILAC than in pulse-SILAC, whereas ER resident proteins appear to have similar prox-SILAC and pulse-SILAC MRF values. This agrees with the vesicular trafficking model: while older proteins are rapidly replaced by nascent proteins in the ER, they are still retained in other subcellular compartments.

In HeLa cells stressed with thapsigardin (Fig. 3E) or SH-SY5Y cells undergoing chemically induced cell differentiation (Fig. 4D, E), while prox-SILAC reveals a global feature of slower protein turnover for ER proteins, the trend is less conspicuous in the whole cell pulse-SILAC dataset (Fig. 3E and Supplementary Fig. 31). However, we note that the difference between prox-SILAC and pulse-SILAC is less striking in SH-SY5Y differentiation experiment than in the case of thapsigargin-induced ER stress in HeLa cells. One reason might be that thapsigargin stress lasts for only 2 h, whereas the differential protocol takes days, allowing sufficient time for the global proteome level to change in response to altered protein trafficking dynamics.

In addition, the observed change in ER prox-SILAC $MRF_{2h}$ values during thapsigargin treatment may have been eclipsed by the whole-cell pulse-SILAC $MRF_{2h}$ values. For example, two heat shock proteins, HSPA5 (from $4 \pm 0\%$ to $15 \pm 1\%$) and HSPA6 (from $4 \pm 0\%$ to $16 \pm 0\%$), have more than 3-fold changes in ER prox-SILAC $MRF_{2h}$, but their corresponding pulse-SILAC $MRF_{2h}$ increase only slightly upon thapsigargin treatment (HSPA5: from $6 \pm 5\%$ to $9 \pm 5\%$; HSPA6: from $3 \pm 1\%$ to $6 \pm 2\%$) (Fig. 3E). These data suggest that while thapsigargin treatment is causing substantial changes in the landscape of newly synthesized proteins within 2 h, the impact has barely reached the global proteome level (i.e. protein abundances).

It is noteworthy that, in addition to their differences in the spatial resolution (subcellular vs. whole-cell), prox-SILAC and pulse-SILAC are also looking at different aspects of protein turnover. For example, unlike pulse-SILAC, prox-SILAC is affected by subcellular protein trafficking. In addition, our data have indicated a bias in APEX2 labeling towards newer proteins within the mitochondrion, possibly due to their increased accessibility as compared to their older counterparts. This aspect, along with other potential APEX2 limitations such as sensitivity towards local pH and redox states, warrants careful considerations when interpreting prox-SILAC data. Nevertheless, despite the increased complexity in result interpretation, prox-SILAC still offers valuable complementary data to pulse-SILAC with discovery potentials.

## Methods

### Molecular cloning
For creating cell lines stably expressing HRP fusion protein, pLX304-SS-HRP-KDEL plasmid and pLX304-SS-HRP-eGFP-KDEL plasmid were constructed through Gibson assembly (Biodragon). The SS-HRP sequence was amplified from plasmid pcDNA3.1-SS-HRP-TM. The KDEL sequence or eGFP-KDEL sequence was fused to the C-terminus of HRP. PCR products were purified by gel extraction kit (Omega). Plasmids were extracted from *E. coli* by EndoFree Mini Plasmid Kit II (Tiangen).

### Cell culture
HEK293T cells, Hela cells or SH-SY5Y cells were cultured in DMEM medium supplemented with 10% fetal bovine serum (FBS) at 37 C° under 5% $CO_2$. To prepare lentivirus, HEK293T cells plated in 6-well plates were transfected at ~70% confluency with Mito-APEX2 plasmid, SS-HRP-KDEL plasmid or SS-HRP-eGFP-KDEL plasmid (1 µg, pLX304 vector), together with two lentivirus plasmids, pCMV-dR8.91 (1 µg) and pVSV-G (0.5 µg) using 7 µL Lipo3000 reagent (Invitrogen) for 6 h. After 48 h, the supernatant containing lentivirus was collected and filtered through a 0.45 µm filter (Sartorius). 1 mL of the lentivirus was added to HEK293T cells, Hela cells or SH-SY5Y cells plated in 6-well plate at ~70% confluency. After 48 h, the culture medium was exchanged to fresh complete medium containing 5 µg/mL blasticidin. Cells were maintained in culture medium containing blasticidin for at least 7 days. Immunofluorescence assay was used to confirm the expression and location of APEX2 or HRP in these cell lines.

To induce ER stress in HeLa cells expressing SS-HRP-KDEL, cells were incubated with complete culture medium (DMEM with 10% FBS) containing 1 µM thapsigargin at 37 °C under 5% $CO_2$. For SH-SY5Y differentiation experiments, cells were seeded with around 30% confluency onto glass slides coated with matrigel (for fluorescence imaging) or 10-cm dish (for LC-MS/MS analysis) and cultured for 24 h. On D0, the cell culture medium was replaced with medium containing 10 µM retinoic acid. The medium was replaced every 2 days. On D7, the medium was replaced with DMEM (without FBS) containing 10 ng/ml brain-derived neurotrophic factor (BDNF).

## Probe synthesis
### Synthesis of biotin-phenol.

**Biotin-NHS**                    **Biotin-Phenol**

Biotin-NHS was prepared by dissolving biotin (4.89 g, 24.0 mmol), N-hydroxysuccinimide (2.30 g, 20.0 mmol) and 1-(3-dimethylaminopropyl)-3-ethylcarbodiimide hydrochloride (4.6 g, 24.0 mmol) to 120 mL warm DMF. The reaction mixture was stirred overnight at room temperature. After the solvent was concentrated to 20 mL under vacuum, pour it into 100 mL cold ethanol. Filter the precipitation to obtain a white solid. The yield is 71%. $^1$H NMR (400 MHz, $d_6$-DMSO): 6.42 (1H, s), 6.36 (1H, s), 4.31 (1H, m), 4.15 (1H, m), 3.11 (1H, m), 2.85 (1H, m), 2.81 (4H, s), 2.67 (2H, t), 2.58 (1H, d), 1.74-1.30 (6H, m).

Biotin-Phenol was prepared by adding biotin-NHS (0.17 mg, 0.5 mmol), tyramine (0.082 g, 0.6 mmol) and triethylamine (210 μL,1.5 mmol) in 15 mL DMF. The reaction mixture was stirred overnight at room temperature. After the solvent was removed under vacuum, the crude mixture was purified by a C18 reverse phase column on semi-preparative UPLC with a gradient of 0 to 60% methanol in water to obtain a white solid. The yield is 65%. $^1$H NMR (400 MHz, $d_6$-DMSO): 9.13 (1H, s), 7.79 (1H, t), 6.97 (2H, d), 6.66 (2H, d), 6.42 (1H, s), 6.35 (1H, s), 4.31 (1H, m), 4.12 (1H, m), 3.18 (2H, dd), 3.08 (1H, m), 2.83 (1H, dd), 2.57 (3H, m), 2.03 (2H, t), 1.65-1.39 (4H, m), 1.35-1.19 (2H, m). The MS characterization was performed on UPLC-MS, calculated for $C_{18}H_{26}N_3O_3S$: $[M + H]^+$ at 364.17 and found at 364.64.

## Proximity labeling and fluorescence microscopy

Cells stably expressing APEX2 or HRP were plated on glass coverslips placed in the wells of a 24-well plate. For proximity-dependent labeling, cells were incubated in 250 μL of 500 μM biotin-phenol probe in complete medium for 30 min at 37 °C. 2.5 μL of freshly prepared 100 mM $H_2O_2$ was added into the medium with brief agitation to achieve a final concentration of 1 mM. After 1-min labeling, the reaction medium was replaced with 500 μL quencher solution (1 mM sodium azide, 1 mM sodium ascorbate and 500 μM Trolox in PBS solution) for 2 min. Then cells were washed again with quencher solution and twice with PBS. Cells were fixed with 4% (w/w) formaldehyde for 15 min at 4 °C.

For immunofluorescence staining, cells were washed with PBS for three times and permeabilized with PBS containing 0.1% (v/v) Tween-20 and 0.2% (v/v) Triton-100 for 15 min at 4 °C. After washing with PBS solution three times, cells were blocked by 5% (w/v) BSA in PBST buffer (PBS buffer containing 0.1% Tween-20). For mito-APEX2 construction, 1:1000 dilution of mouse anti-TOMM20 (abcam) and rabbit anti-V5 (abcam) were incubated for 1 h at room temperature, followed by goat anti-mouse Alexa Flour 568 (Thermo Fisher Scientific), goat anti-rabbit Alexa Flour 488 (Thermo Fisher Scientific), streptavidin conjugated Alexa Fluor 647 dye (Thermo Fisher Scientific) and DAPI (all 1:1000) as secondary antibody. For HRP-KDEL construction, 1:1000 dilution of rabbit anti-calnexin (abcam) and mouse anti-HA (biodragoon) were incubated for 1 h at room temperature, followed by goat anti-mouse Alexa Flour 488 (Thermo Fisher Scientific), goat anti-rabbit Alexa Flour 568 (Thermo Fisher Scientific), streptavidin conjugated Alexa Fluor 647 dye and DAPI (all 1:1000) as secondary antibody.

Stained cells were imaged on an inverted fluorescence microscope (Nikon-TiE) equipped with a spinning disk confocal unit (Yokogawa CSU-X1) and a scientific complementary metal-oxide semiconductor camera (Hamamatsu ORCA-Flash 4.0 v.2). This system was controlled with a customized software written in LabVIEW v.15.0 (National Instruments).

## Prox-SILAC labeling in cells

For SILAC medium preparation, L-arginine and L-lysine were replaced by heavy-labeled L-arginine-$^{13}C_6$-$^{15}N_4$ and heavy-labeled L-lysine-$^{13}C_6$-$^{15}N_2$. Each L-arginine-$^{13}C_6$-$^{15}N_4$ introduces +10.0083 Da mass difference in a tryptic digested peptide, and the L-lysine-$^{13}C_6$-$^{15}N_2$ introduces +8.0142 Da mass difference respectively. The 1000x concentrated stock solutions with L-arginine-$^{13}C_6$-$^{15}N_4$ (88.8 mg/mL), L-lysine-$^{13}C_6$-$^{15}N_2$ (154 mg/mL) in PBS was filtered by 0.22-um syringe filter and stored at 4 °C. The pulse-SILAC medium was generated by adding the heavy-labeled amino acid stock into SILAC DMEM with 10% SILAC fetal calf serum. Cells were cultured in normal DMEM medium for 5–7 generation for pro-SILAC experiments. The culture medium was replaced with the pulse-SILAC medium for the indicated duration. For instance, when measuring $MRF_{4h}$ of proteins, cells were incubated in pulse-SILAC medium for 3.5 h, and then the medium was replaced with pulse-SILAC medium containing 500 μM biotin-phenol probe for another 30 min. APEX labeling was triggered by adding 100x $H_2O_2$ solution (100 mM) into cell culture with gentle rocking. After 1-min labeling, the reaction medium was replaced with 500 μL quencher solution (1 mM sodium azide, 1 mM sodium ascorbate and 500 μM Trolox in PBS solution) for 2 min. Then the cells were washed again with quencher solution and twice with PBS.

For cell surface protein labeling, HEK293T-SS-HRP-KDEL cells were incubated in pulse-SILAC medium for 2 h. Then the medium was exchange to HBSS buffer containing 0.5 mg/mL sulfo-NHS-LC-biotin reagent for 2 min. The cells were wash with PBS buffer for three times

Cells were lysed with RIPA buffer containing 25 mM Tris•HCl pH 7.6, 150 mM NaCl, 1% NP-40, 1% sodium deoxycholate, 2% SDS and protease inhibitors cocktail for 15 min at 4 °C. The high concentration of SDS facilitates the extraction of membrane proteins. Cells were scraped and lysed via ultrasonication on ice bath. The lysate was centrifuged at 20000 g at 4 °C for 10 min. For western blotting analysis, the collected supernatant was mixed with 5X loading buffer and heated at 95 °C for 10 min. For subsequent proteomic experiments, protein samples were purified by cold methanol precipitation at −80 °C for 3 h.

## Western blotting analysis

Protein samples were separated by 10% SDS-PAGE gel under 110 V for 70 min and transferred to PVDF membrane (Bio-Rad) under 240 mA for 70 min. The blots were blocked with 5% BSA in TBST solution (TBS buffer, pH ~7.4, containing 0.1% Tween-20) overnight at 4 °C. The blots were then immersed with 0.25 μg/mL streptavidin-HRP (Thermo Fisher Scientific) at room temperature for 1 h. For detecting the expressing of fusion V5 or HA epitope tag, the blots were incubated with mouse anti-V5 monoclonal antibody (Biodragon, 1:5000) or mouse anti-HA monoclonal antibody (Biodragon, 1:5000) as primary antibody at room temperature for 1 h, followed by HRP-conjugated goat anti mouse IgG as secondary antibody for 1 h. At last, the blots were washed with TBST solution three times, developed with Clarity Western ECL substrate (Bio-Rad) and imaged by a Chemidoc imager (Bio-Rad).

## Enrichment of biotinylated proteins and samples preparation for MS analysis

Purified proteins were resolubilized in 0.5% (w/v) SDS aqueous solution and quantified via BCA protein assay. 5 mg proteins were incubated with 250 μL streptavidin beads for 3 hours at room temperature with gentle rotation. Then the beads were washed twice with 2% (w/v) SDS aqueous solution, twice with 8 M urea, and twice with 2 M sodium chloride. Proteins on the beads were incubated with 6 M urea and 10 mM dithiothreitol for 15 min, and subsequently alkylated with 20 mM iodoacetamide in the dark at 35 °C for 30 min. After washing twice with triethylammonium bicarbonate buffer, proteins on beads were treated with 4 μg trypsin (Promega) for 16 hours at 37 °C. A small fraction of input, supernatant, flow through, and elute sample (biotin competition under 95 °C) were kept for further western blotting analysis. Thereafter, released peptides were collected from the supernatant following centrifugation at 15000 g for 10 min. The digested peptides were fractionated using high pH reversed-phase peptide fractionation kit. After loading on the column, the samples were eluted by 5%, 7.5%, 10%, 12.5%, 15%, 17.5%, 20%, 50% acetonitrile solution in 0.1% triethylamine respectively. Finally, the eluted samples were mixed pairwise (the nth and n+4th fractions) to yield 4 fractions for LC-MS/MS analysis.

## Protein digestion in solution

Purified proteins were resolubilized in 8 M urea solution (urea dissolved in 50 mM Tris-HCl, pH ~ 8.5) and quantified via BCA protein assay (about 200 μL to 1 mL urea for 1 mg protein). 0.5 M fresh dithiothreitol solution was added to a final concentration of 10 mM and incubated at 55 °C for 30 min. Then 0.5 M iodoacetamide solution was added to a final concentration of 30 mM and incubated at 37 °C for another 30 min in the dark. Thereafter, excess iodoacetamide was neutralized with additional dithiothreitol (final concentration of 20 mM) at 55 °C for 15 min. The urea concentration was then diluted by adding 7X volume of 50 mM Tris-HCl (pH ~ 8.5). CaCl$_2$ solution (final concentration of 1 mM) was added to facilitate the digestion as well. Proteins were digested with trypsin (sigma) at a 1:20 ratio at 37 °C for more than 16 h. Digested peptides were fractionated using high pH reversed-phase peptide fractionation kit as aforementioned.

## LC-MS/MS analysis

Peptides were separated using a loading column (100 μm × 2 cm) and a C18 separating capillary column (75 μm × 15 cm) packed in-house with Luna 3 μm C18(2) bulk packing material (Phenomenex, USA). The mobile phases (A: water with 0.1% formic acid and B: 80% acetonitrile with 0.1% formic acid) were driven and controlled by a Dionex Ultimate 3000 RPLC nano system (Thermo Fisher Scientific). The LC gradient was held at 2% for the first 8 min of the analysis, followed by an increase from 2% to 10% B from 8 to 9 min, an increase from 10% to 44% B from 9 to 123 min, and an increase from 44% to 99% B from 123 to 128 min.

For the samples analyzed by Orbitrap Fusion LUMOS Tribrid Mass Spectrometer, the precursors were ionized using an EASY-Spray ionization source (Thermo Fisher Scientific) source held at +2.0 kV compared to ground, and the inlet capillary temperature was held at 320 °C. Survey scans of peptide precursors were collected in the Orbitrap from 350-1600 Th with an AGC target of 400,000, a maximum injection time of 50 ms, RF lens at 30%, and a resolution of 60,000 at 200 m/z. Monoisotopic precursor selection was enabled for peptide isotopic distributions, precursors of z = 2–7 were selected for data-dependent MS/MS scans for 3 seconds of cycle time, and dynamic exclusion was set to 15 s with a ± 10 ppm window set around the precursor monoisotope.

## Protein identification and quantitation

Raw data files were searched against *Homo sapiens* Uniprot database (downloaded on Nov 19th, 2020). Database search were performed with MaxQuant software (Version 1.6.2.3). The digestion mode was set to trypsin. The maximum number of modifications allowed per peptide was set to five. The maximum number of missed cleavages allowed per peptide was two. Mass shifts of +57.0214 Da (carbamidomethylation, C) was searched as fixed modifications; +15.9949 Da (oxidation, M), +42.0105 Da (acetyl N-terminal) were searched as variable modifications. Mass shifts of +8.0142 Da (Lys8, K) and +10.0083 Da (Arg10, R) were set as heavy label. The FDR were set to <1%. The mass spectrometry proteomics data have been deposited to the ProteomeXchange Consortium via the PRIDE[60] partner repository with the dataset identifier PXD037569.

For each identified and quantified protein, its metabolic replacement fraction (MRF) was calculated from its SILAC H/L ratio, as follows:

$$MRF = \frac{\text{Intensity}_{heavy}}{\text{Intensity}_{heavy} + \text{Intensity}_{light}} = \frac{H/L\text{ratio}}{1 + H/L\text{ratio}} \tag{1}$$

Only the proteins identified in both replicates were taken for subsequent analysis. Their metabolic replacement fractions were the arithmetic mean values of the two replicates.

The half-life of protein was calculated by fitting the following exponential equation with the MRF values identified at 1 h, 2 h, 4 h, 8 h and 12 h.

$$MRF = 1 - e^{-kt} \tag{2}$$

$$t_{1/2} = \frac{\ln(2)}{k} \tag{3}$$

## Gene Ontology analysis

To calculate the mitochondrial specificity of our proteome, the mitochondrial proteins were confirmed by searching for "mitochon" in the GO terms from the Uniprot-GO annotations. For secretory pathway specificity, proteins with "endoplasmic reticulum", "Golgi", "plasma membrane", "endosome", "lysosome", "nuclear envelop", "nuclear membrane", "perinuclear region of cytoplasm", "extracellular space" or "vesicle" descriptions in Uniprot-GO annotation were identified as secretory pathway proteins. Further, the secretory proteins with only "endoplasmic reticulum" annotation were regarded as "ER residents". The secretory proteins without "endoplasmic reticulum" annotation were described as "post-ER proteins".

## Statistics and reproducibility

Immunofluorescence images are acquired from at least three fields of view. All proteomic experiments contain at least two biological replicates. Statistical analysis (e.g. Pearson's r) of the proteomic data was performed with GraphPad Prism 9 and Excel under a two-sided manner.

## Reporting summary

Further information on research design is available in the Nature Portfolio Reporting Summary linked to this article.

# Data availability

The mass spectrometry proteomics data have been deposited to the ProteomeXchange Consortium via the PRIDE partner repository with the dataset identifier PXD037569. The PDB entries used in this work have the following IDs 3J9M, 6VAX, 5Z62. The raw data of uncropped gel and bar charts are provided in the Source Data file. Source data are provided with this paper.

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

## Acknowledgements

We thank L. Peng for assistance with image analysis, T. Yue for assistance with probe synthesis, and all lab members for helpful discussions. This work was supported by the Ministry of Science and Technology (2022YFA1304700, 2018YFA0507600), the National Natural Science Foundation of China (32088101). PZ is sponsored by Bayer Investigator Award. We thank the Analytical Instrumentation Center in Peking University for assistance with MS sample identification and Ms. W. Zhou for help with MS results analysis.

## Author contributions

P.Z. conceived the project. F.Y. and P.Z. designed experiments. F.Y., Y.L., X.Z. and P.M. performed experiments. F.Y., Y.L. and P.Z. analyzed the data and wrote the paper.

## Competing interests

The authors declare no competing interests.
