## [Peer Review File · Nature Communications]

Spatially resolved mapping of proteome turnover dynamics with subcellular precisionREVIEWER COMMENTS

Reviewer #1 (Remarks to the Author):

In the submitted manuscript, the authors introduced interesting techniques combining proximity labeling and pulse-SILAC. They revealed heterogeneous protein turnover in mitochondria and ER. Further, they identified subsets of proteins in the differentiation of SH-SY5Y cells. The authors have generated interesting proteome datasets, however, validation and implication of data should be investigated considering the broad readership of Nature Communications.

1) In mitochondrial matrix data, TR value is variable from 11 to 79%. Even in the same mitochondrial ribosome complex, TR values of each protein are very different even though neighboring proteins tend to show similar values. I wonder that, considering a functional biological unit is one complex, why does each component protein of the same complex show a very different turnover ratio? and, why mitochondria turnover 80% of NDUFB8 in just 4 hours although synthesizing new peptides utilizes much more ATP or energy. Is it related to oxidative damage or protein degradation? They wrote in the abstract, their analysis shed light on the "mechanism of hierarchical assembly", but it is not sufficiently addressed. The authors should address this point with experimental evidence or detailed discussion.

2) In ER data, authors claim that trafficking affects the turnover ratio. If authors analyze secreted proteins in culture supernatant labeled by prox-SILAC, it can support their conclusion. and, using protein synthesis inhibitors such as cycloheximide can further validate their hypothesis.

Minor points:

1) The authors mentioned "While the above tools are powerful for analyzing proteins turnover, they are only applicable to the whole-cell proteome level and lack subcellular spatial resolution (page 2)", however, this sentence needs to be corrected because mitochondrial and ER protein information can be extracted from pulse-SILAC data by using previously annotated subcellular protein information (as shown in in Fig. 1C and Fig. 2B) although some level of ambiguity has remained. Using this approach, pulse-SILAC has already resolved the assembly kinetics of mitochondrial respiratory complexes [REF 15] and mitochondrial ribosome [REF 16] as commented by authors in Page 2. Thus, I'm wondering what is the big advantage of prox-SILAC over pulse-SILAC x subcell information approach because currently prox-SILAC method shows less coverage of mito and ER proteins than using pulse-SILAC x subcell information approach (Fig. 1C and Fig. 2B). The authors also need to check whether measured turnover rates of mitochondrial respiratory complexes [REF 15] and mitochondrial ribosome [REF 16] have correlation between prox-SILAC data and pulse-SILAC data. If some subunit has significantly different turnover rate between two datasets, this might be an interesting point that can be highlighted and further discussed in the manuscript.

2) Currently, prox-SILAC method shows less coverage of mitochondrial matrix proteins (~183 proteins) than original report using mito-APEX and SILAC method (~495 proteins) [PMID: 23371551]. It would be great if the authors could clarify this issue in the Discussion part.

3) In supplementary methods - Cell culture The exact plasmid name of Delta8.91 should be written. (pCMV-dR8.91?) 0.45 uM should be 0.45 μ M.

4) Supplementary table 2,3,4 - Column definitions The turnover rate should be the turnover ratio as the main text. TR should be $H/L / (1 + H/L)$ as the main text.

5) In Fig.1c and 2b pulse-SILAC should be pulse-SILAC. (Typo)

Reviewer #2 (Remarks to the Author):

This manuscript describes a novel application for measuring protein turnover at a subcellular level. While none of the components themselves are new, the combination and resulting information that

they obtain is novel and interesting.

Concerns:

How to express and what to call protein turnover is not universally agreed upon. "Turnover", as expressed here, suggests that the overall level of a protein does not change over the course of the experiment (i.e., assumes steady state) but I do not believe that the authors measured the change in expression at their timepoints (4, 8, 12 h). Other variations of turnover involve measuring synthesis and degradation (e.g., in PMID 24045637). Some discussion of this caveat should be added. Or, if the authors have measured total protein changes then their TR values should be expressed relative to this

The biggest issue with this manuscript is that there is only two replicates of each experiment done and, thus, no statistics to back up some of the biological claims. While I do not like requesting large extra experiments unless truly necessary, in this case I feel that the claims would be very substantially strengthened.

How is it possible to have a negative change in the TR value for a given protein from an early timepoint to a later timepoint? This seems to happen for a large fraction of proteins (and is visible in Fig. 1e) but does not seem biologically possible. The authors need to address what could be going on here

Significant figures in error values. What does it mean that you could measure error to three significant figures? E.g., $56.9 \pm 15.8\%$ on page 4. This is not appropriate and is done throughout the manuscript. The values I have cited here would more appropriately be expressed as $57 \pm 16\%$

Reviewer #3 (Remarks to the Author):

Feng Yuan and collaborators combine proximity-dependent protein labeling with enrichment, metabolic labelling and proteomics in cell culture to provide subcellular-level measures of protein turnover.

The technical application of proximity-dependent enrichment and SILAC labelling is relatively little explored and could be potentially useful for this field. The topic of how sub-cellular localization influences protein stability/turnover is extremely interesting.

Unfortunately, I have major concerns about the actual relevance of the work since the authors do not properly showcase the usefulness of their strategy. I see two major purposes for combining proximity-dependent enrichment and SILAC labelling. The first one is that, using this approach, organelle specific sub-proteomes (that cannot be normally detected) become accessible. The second one is that looking at the subcellular proteome vs. the total proteome some important changes can be observed in the labelling efficiency. The main problem of this manuscript is that the authors consider only very superficially these two aspects.

To address these major concerns, the authors should systematically quantify protein lifetimes/labelling profiles across sub cellular proteomes, compare them to the total proteome, report proteins that cannot be seen in the overall proteome and show that indeed the localization of proteins can influence the relative age/labelling profile of sub cellular proteomes.

Major

1) Fig. S1: which should be the enriched fraction? The eluted sample? From panel A it does not seem to be the case and it looks like the enrichment did not work at 4h. Could the authors comment on that?

2) A quick GO over-representation analysis of the 162 proteins that were identified and quantified across different replicates (end of page 3) reveals that beside the mitochondrial matrix there are several proteins that belong to the nucleoid and to other compartments. This raises the question of

how precise is the enrichment method applied. This is particularly relevant in the light of what the authors write in their introduction: "However, organelle purification methods are prone to introducing contamination and are not generally applicable to other sub-cellular locations, which motivates us to explore alternative methods for studying sub-cellular proteome turnover." How much quantitatively enriched are mitochondrial proteins using their approach vs other proteins? A measure of relative enrichment vs initial abundance should be implemented in the analysis to facilitate the interpretation of the results. If possible, a comparison with other sub cellular fractionation methods should also be implemented, to showcase the usefulness of the approach proposed here. One other angle could be to showcase proteins that are not detected by normal MS in the pulse SILAC but that can be observed following enrichment (and thus deeper coverage of that sub cellular proteome). Similar to the mitochondrial labelling, the other enrichment strategies should be compared to other available fractionation methods.

3) The turnover ratio (%) used across the manuscript (introduced in the upper part of Page 4), is in this work a measure of the metabolic labelling. When expressed to a percentage actually corresponds to the Heavy fraction (%). To avoid misunderstandings, since proteins generally get labeled with exponential dynamics, this measure should be renamed to "normalized heavy fraction" since a measure of turnover in this field often indicates a certain measure of time (a turnover rate) and the term could be misleading. Have the authors considered to measure the half-life (the lifetimes) for the proteins analyzed?

4) It is obvious that longer pulses would correspond to higher metabolic labelling. Likewise, differences in protein lifetimes in mitochondrial proteins (and in mitochondrial ribosomes) have been previously observed with no need of proximity labelling enrichment. The relevant question here is what is the difference in the metabolic labelling of the mitochondrially-enriched proteins vs. the labelling of non-enriched proteins. Do the authors see a difference? This aspect should be addressed systematically taking into account all labelling time points and enriched vs. not-enriched proteomes. As also mentioned on point 1, one other possibility, if there are no differences would be to show that the proximity labelling has other advantages (such as deeper coverage of the proteome). The advantages of this approach should be more clearly presented in the work.

5) Although it is tantalizing to speculate that the secretory pathway shows a specific clearance of older proteins that would be trafficked to the Golgi (end of page 6), it cannot be excluded that faster labelling in the ER simply reflects an overall faster turnover of all ER proteins vs mitochondrial proteins (as previously reported in several occasions in vitro and in vivo). The point of how sub-cellular localization influences protein turnover should be the most important aspect of this work. In order to address this the authors should do proximity labelling in different organelles and compare the labeling across sub-proteomes and with the overall proteome (see below). In this context, what is the definition of secreted proteins for the authors? Proteins exposed to the plasma membrane or actually "strictly secreted" (e.g. hormones) and/or cleaved proteins such as APP? Strictly secreted proteins are lost in the medium upon secretion and proteolytic cleavage and their labelling profile will reflect this process. In other words, their lifetimes will be extremely short (so higher relative H labeling as in reality the only thing that is measured is how long they stay within the cell, not how long they actually live). This observation, has not much to do with the fact that proteins trafficked to the Golgi are having a shorter lifetime. Moreover, the difference might be explained by evolutionary constraints that render some proteins more or less prone to damage or degradation. For this type of analysis proteins from the plasma membrane (that are not secreted) could be taken as a proxy and their relative labeling could be measured in the ER, in the Golgi and in the plasma membrane upon sub-cellular fractionation or surface labelling or, if the authors wish, using complementary proximity labelling approaches. The authors should address comparatively if at the same labelling time point the protein localization influences the labelling efficiency. The authors should refrain from using a scheme (Fig. 2G) in support of their hypothesis.

6) Are changes observed in Fig. 3E and Fig. 4E specific for the ER-enriched proteome or these changes are general (and could be thus observed also in the absence of proximity labelling and enrichment)? The authors should address this point and showcase the usefulness of their method and why looking at this sub-cellular proteome is relevant for studying these issues.

Minor

- 0) For making comments on specific text sections it would be really useful to have line numbers.
- 1) The introduction does focus predominantly on in vitro cellular work, although great progress has been made in the protein turnover field in-vivo (in whole animals). Since the analysis of protein lifetimes in culture might reflect a very limited representation of the situation in vivo, the authors should at least acknowledge that literature that literature in the introduction.
- 2) There is at least an exception to the statement "While these methods have been broadly applied to profile the abundance of proteins within subcellular structures they have not been used for investigating the turnover dynamics of subcellular proteome.". There is one work using MIMS (NanoSIMS) combining APEX and metabolic labelling: <https://pubmed.ncbi.nlm.nih.gov/31719114/> whic should be probably cited in this context. The authors should also check more carefully if there are other such examples in the literature and acknowledge them.
- 3) The quality of Fig s15 I not acceptable (the image is completely blurred / out of focus).

We thank all three reviewers for their thoughtful comments to help us improve this manuscript. In the revised manuscript, we have provided additional experimental data of replicated prox-SILAC on the cell surface and replicated pulse-SILAC in mitochondrial matrix and ER lumen, in stressed conditions and during chemically induced differentiation. As suggested by the reviewers, we have also added discussions to compare our findings with previous reports, including pulse-SILAC analysis of purified mitochondria. Please see our point-by-point response below. In the revised manuscript, these changes have been marked in red.

Reviewer #1 (Remarks to the Author):

In the submitted manuscript, the authors introduced interesting techniques combining proximity labeling and pulse-SILAC. They revealed heterogeneous protein turnover in mitochondria and ER. Further, they identified subsets of proteins in the differentiation of SH-SY5Y cells. The authors have generated interesting proteome datasets, however, validation and implication of data should be investigated considering the broad readership of Nature Communications.

1) In mitochondrial matrix data, TR value is variable from 11 to 79%. Even in the same mitochondrial ribosome complex, TR values of each protein are very different even though neighboring proteins tend to show similar values. I wonder that, considering a functional biological unit is one complex, why does each component protein of the same complex show a very different turnover ratio? and, why mitochondria turnover 80% of NDUFB8 in just 4 hours although synthesizing new peptides utilizes much more ATP or energy. Is it related to oxidative damage or protein degradation? They wrote in the abstract, their analysis shed light on the "mechanism of hierarchical assembly", but it is not sufficiently addressed. The authors should address this point with experimental evidence or detailed discussion.

Response: We thank the reviewer for raising these interesting questions. Indeed, we are also intrigued and perplexed by the large variations in the TR values even within the same protein complexes. We note that we are not the first to report this phenomenon. Similar trends have been observed for mito-ribosomal proteins (Bogenhagen DF et al, PubMed ID: 29444443) and respiratory complexes (Bogenhagen DF and Haley JD, PubMed ID: 31974161) using SILAC pulse-labeling in HeLa cell. In these studies, the H:L ratios differ by as much as 5-fold for ribosomal proteins and 4-fold for respiratory complex proteins, indicating that these proteins are "synthesized and imported in considerable excess over the amount required to support assembly", as reasoned by the authors. Our data are consistent with the previous observations, which, together, may not necessarily contradict with the view that "a

functional biological unit is one complex” as the reviewer pointed out, but rather suggest a complicated assembly process that warrants further investigation.

We share with the reviewer’s notion that cells would consume excessive amount of energy to over-synthesize the peptides. Thus, there should be some benefits in doing so. We agree with the reviewer’s suggestion that oxidative damage may play a role. As subunits in the OXPHOS machinery are liable to be oxidized and need to be replaced, excessive copies in the mitochondria could act as a reserve that is necessary to keep the OXPHOS machinery functioning, thus leading to higher turnover for a subset of mitochondrial proteins. The evidence supporting this hypothesis is that the TR values (MRF_{8h}, metabolic replacement fraction at 8 hr, in the revised manuscript) of respiratory complex I, which involves electron transport, are overall much higher than those in complex V, which does not involve electron transport (mean values 57% vs. 35%). In the revised manuscript, we have added the above discussions on page 16:

“Notably, remarkable variations in the MRF values of protein subunits within mitochondrial ribosome complexes and respiratory complexes have also been reported in previous work via pulse-SILAC labeling in HeLa cells^[17]. In these studies, the H/L ratios exhibited a difference of 5-fold for ribosomal proteins and 4-fold for respiratory complex proteins, indicating that these proteins are synthesized and imported into mitochondria with considerable excess over the minimum amount required to support assembly. The above observation revealed a complicated assembly process that warrants further investigation. As OXPHOS components are liable to be oxidized by leaked electrons from electron transfer chain, excessive copies in the mitochondria may act as a reserve to quickly replace the damaged subunits, thus keeping the OXPHOS machinery functioning smoothly.”

2) In ER data, authors claim that trafficking affects the turnover ratio. If authors analyze secreted proteins in culture supernatant labeled by prox-SILAC, it can support their conclusion. and, using protein synthesis inhibitors such as cycloheximide can further validate their hypothesis.

Response: We agree with the reviewer that analyzing the turnover of secreted proteins could support our ER prox-SILAC dataset. However, our attempts to enrich secreted SILAC-encoded proteins from the cell culture medium failed, largely due to both the low amount of secretion and the high background interference from serum proteins. As an alternative, we labeled the cell surface proteome with membrane-impermeant biotin-sulfo-NHS reagent and enriched SILAC-encoded proteins through streptavidin-coated beads. Although these proteins are not secreted to the extracellular media, they also need to traffic through the secretory pathway from the ER to the cell surface. Thus, it would be interesting to compare the TR values of the same protein

cohort when they are in the ER versus when they are on the cell surface.

The overlap of our cell surface pulse-SILAC proteome and ER prox-SILAC proteome yielded 34 proteins. As expected, our data reveal substantially higher TR values (MRF values in the revised manuscript) in the ER lumen versus on the cell surface. Examples include membrane receptors, such as TFRC (71±1% vs. 4±0%) and IGF2R (55±3% vs. 5±3%), and ion pump proteins, such as ATP1A1 (68±1% vs. 19±0%). We attributed these differences to the rapid clearance of membrane proteins from the ER and the delay of their appearance on the cell surface due to the time required for protein maturation and vesicular trafficking.

Interestingly, we also observed a reversed trend (i.e. low TR in the ER and high TR on the cell surface) for a few proteins, namely SERPINH1 (5.0±0% vs. 86±0%) and PDIA6 (4±5% vs. 56±16%). Both are protein chaperones in the ER lumen. We note that PDIA6 is also reported to associate with other substrate on the cell surface, in addition to its major function in the ER (PMID: 25624318). We thus speculate that the higher MRF values of these protein chaperones on the cell surface suggest the presence of a sub-population with distinct roles, although we currently do not have experimental proof to support this claim. In the revised manuscript, we have supplemented the above results and discussion on page 9:

“To further investigate protein turnover on the plasma membrane, we performed 2-hour pulse-SILAC labeling in HEK293T cells and labeled the cell surface protein population with the membrane-impermeant chemical reagent, Sulfo-NHS-LC-Biotin. Similar to our workflow of prox-SILAC, biotinylated proteins were subsequently enriched with streptavidin-coated beads and analyzed by LC-MS/MS to derive the MRF_{2h} values (Figure S18). In two replicated experiments, we have identified and quantified 508 cell surface proteins (Figure S19). For the 34 proteins identified in both ER lumen and cell surface prox-SILAC datasets, their MRF_{2h} values differ substantially (Figure S20). Membrane receptors and ion pumps such as TFRC (71±1% vs. 4±0%), ATP1A1 (68±1% vs. 19±0%) and IGF2R (55±3% vs. 5±3%), exhibit much higher ER lumen MRF values, which is attributed to the time required for them to traffic from the ER lumen to the plasma membrane. To our surprise, two protein chaperones, SERPINH1 (5.0±0% vs. 86±0%) and PDIA6 (4±5% vs. 56±16%) exhibit higher cell surface MRF_{2h} than ER MRF_{2h} values. SERPINH1 (Hsp47) is a collagen-specific molecular chaperone and plays an important role in the collagen biosynthesis^[35]. While PDIA6 mainly functions as a chaperone that inhibits aggregation of misfolded proteins during unfolded protein response (UPR)^[36], it has also been reported to bind integrin β3 subunit on the cell surface to promote platelet activation after stimulation^[37]. We thus speculate that the higher MRF values of protein chaperones may be associated with their distinct functions on the cell surface. Together, the above comparison of prox-SILAC at various subcellular compartments highlight its advantage in investigating

spatially resolved protein turnover.”

Figure S18. Experimental scheme of pulse-SILAC in HEK293T-SS-HRP-KDEL cells and cell surface protein enrichment. To tag the nascent proteins, HEK293T-SS-HRP-KDEL cells were briefly cultured in heavy SILAC medium for 2 hours. To label cell surface proteome, cells were incubated with sulfo-NHS-LC-biotin for 10 mins to capture the cell surface proteins. The nascent PM proteins were thus doubly labeled with both biotin and heavy arginine/lysine. Following cell lysis, biotinylated proteins were enriched by streptavidin-coated beads. The metabolic replacement fractions of targeted proteins were identified by subsequent LC-MS/MS analysis.

Figure S19. Pulse-SILAC experiments on cell surface. (A) Venn diagram showing the overlap of the two cell surface pulse-SILAC replicates. (B) Scatter plot showing the MRF values of the two replicates.

Figure S20. Scatter plot of MRF_{2h} values of secretory pathway proteins measured in ER-prox-SILAC against cell surface pulse-SILAC.

We also thank the reviewer for suggesting cycloheximide (CHX) treatment to shut down protein synthesis. We considered this approach but realized that the heavy isotope-labeled amino acids would not get incorporated into newly synthesized polypeptides when cells were treated with CHX, which would result in zero MRF values that might not provide meaningful information. We also realize that the term TR (turnover ratio) may mislead the readers, as *protein turnover* often refers to the replacement of older proteins as they are broken down within the cell, whereas our measurement of the prox-SILAC H/L ratio reflects the combined effects of protein synthesis, degradation and local trafficking in specific subcellular localization. Thus, in the revised manuscript, we have changed the name of *turnover ratio* to *metabolic replacement fraction (MRF)*. We have also added the following discussion to caution the readers of this definition and its difference from conventional protein turnover.

On page 16: “Conventional pulse-SILAC labeling on the whole cell level reveals the protein turnover dynamics, which includes protein synthesis and degradation. Notably, the MRF values measured with prox-SILAC reflects the combine effects of not only protein synthesis and degradation, but also local trafficking in specific subcellular localization.”

Minor points:

1) The authors mentioned "While the above tools are powerful for analyzing proteins turnover, they are only applicable to the whole-cell proteome level and lack subcellular spatial resolution (page 2)", however, this sentence needs to be corrected because mitochondrial and ER protein information can be extracted from pulse-SILAC data by using previously annotated subcellular protein information (as shown in in Fig. 1C and Fig. 2B) although some level of ambiguity has remained. Using this approach, pulse-SILAC has already resolved the assembly kinetics of mitochondrial respiratory complexes [REF 15] and mitochondrial ribosome [REF 16] as commented by authors in Page 2. Thus, I'm wondering what is the big advantage of prox-SILAC over pulse-SILAC x subcell information approach because currently prox-SILAC method shows less coverage of mito and ER proteins than using pulse-SILAC x subcell information approach (Fig. 1C and Fig. 2B). The authors also need to check whether measured turnover rates of mitochondrial respiratory complexes [REF 15] and mitochondrial ribosome [REF 16] have correlation between prox-SILAC data and pulse-SILAC data. If some subunit has significantly different turnover rate between two datasets, this might be an interesting point that can be highlighted and further discussed in the manuscript.

Response: We thank the reviewer for pointing out the alternative approach of crossing pulse-SILAC data with subcellular annotation, which should be compared with prox-

SILAC to highlight the unique advantage of our method. Indeed, as the reviewer pointed out, prox-SILAC appears to have lower protein coverage than the pulse-SILAC x subcell information approach (Figure 1C and Figure 2B). However, we note that for proteins with multiple subcellular localizations, pulse-SILAC only measures the overall turnover at the whole cell level, but does not distinguish the differences in turnover within specific subcellular compartments. This is particularly evident for ER lumen-localized proteins. In Figure 2F, we compared ER prox-SILAC against ER pulse-SILAC, revealing that trafficking proteins overall have substantially higher MRF values in prox-SILAC than in pulse-SILAC, whereas ER resident proteins appear to have similar prox-SILAC and pulse-SILAC MRF values. This agrees with the vesicular trafficking model: while older proteins are rapidly replaced by nascent proteins in the ER, they are still retained in other subcellular compartments.

To further support the above argument, we have provided additional experimental data on pulse-SILAC during ER stress or cell differentiation in the revised manuscript. While prox-SILAC shows a global trend of slower protein turnover for ER proteins under thapsigardin-induced stress (Figure 3E) or during chemically induced cell differentiation (Figure 4D-E), the trend is less conspicuous in the whole cell pulse-SILAC dataset (Figures 3E and S31).

Another advantage of prox-SILAC is the capability of detecting low abundance proteins, which are tagged and enriched during affinity purification. These proteins may be otherwise undetectable by LC-MS/MS, due to interference from highly abundant protein background. For example, our mitochondrial prox-SILAC dataset contains 72 proteins that are missed by mito pulse-SILAC, despite the overall larger coverage of the latter (407 mito proteins vs. 162). Similarly, our ER-lumen prox-SILAC datasets in HEK293T cells, HeLa cells, and TG-stressed HeLa cells have uniquely identified 94, 186, and 172 proteins, respectively, as compared to the pulse-SILAC counterparts. With our newly added pulse-SILAC datasets for SH-SY5Y cell differentiation, we confirm that 262, 180, and 90 proteins are uniquely quantified in ER lumen prox-SILAC at D0, D7, and D10, respectively, as compared to the corresponding pulse-SILAC datasets.

In the revised manuscript, we have provided additional discussions on the advantages of prox-SILAC in terms of spatial specificity and coverage (page 18):

“For all the above prox-SILAC experiments, we have performed the equivalent pulse-SILAC experiments at the whole-cell level. Prox-SILAC offers unique coverage of the subcellular proteome as compared with pulse-SILAC. For example, our mitochondrial prox-SILAC dataset contains 72 proteins that are missed by mito pulse-SILAC, despite the overall larger coverage of the latter (407 mito proteins vs. 162). Similarly, our ER-lumen prox-SILAC datasets in HEK293T cells, HeLa cells, and TG-stressed HeLa cells have uniquely identified 94, 186, and 172 proteins, respectively,

as compared to the pulse-SILAC counterparts. With our newly added pulse-SILAC datasets for SH-SY5Y cell differentiation, we confirm that 262, 180, and 90 proteins are uniquely quantified in ER lumen prox-SILAC at D0, D7, and D10, respectively, as compared to the corresponding pulse-SILAC datasets.”

“Prox-SILAC also provides the high spatial specificity required for resolving subcellular MRF values. A comparison between our ER lumen prox-SILAC versus ER lumen pulse-SILAC datasets reveals that trafficking proteins overall have substantially higher MRF values in prox-SILAC than in pulse-SILAC, whereas ER resident proteins appear to have similar prox-SILAC and pulse-SILAC MRF values. This agrees with the vesicular trafficking model: while older proteins are rapidly replaced by nascent proteins in the ER, they are still retained in other subcellular compartments.”

“In HeLa cells under thapsigardin-induced stress (Figure 3E) or SH-SY5Y cells undergoing chemically induced cell differentiation (Figure 4D-E), while prox-SILAC reveals a global feature of slower protein turnover for ER proteins, the trend is less conspicuous in the whole cell pulse-SILAC dataset (Figures 3E and 31).”

“The observed change in ER prox-SILAC MRF_{2h} values during TG treatment may have been eclipsed by the whole-cell pulse-SILAC MRF_{2h} values. For example, two heat shock proteins, HSPA5 (from 4±0% to 15±1%) and HSPA6 (from 4±0% to 16±0%), have more than 3-fold changes in ER prox-SILAC MRF_{2h} but their corresponding pulse-SILAC MRF_{2h} increased only slightly upon TG treatment (HSPA5: from 6±5% to 9±5%; HSPA6: from 3±1% to 6±2%) (Figure 3E). While TG treatment is causing substantial changes in the landscape of newly synthesized proteins within 2 hours, the impact has barely reached the global proteome level (i.e. protein abundances).”

“However, we note that the difference between prox-SILAC and pulse-SILAC is less striking in SH-SY5Y differentiation experiment than in the case of TG-induced ER stress in HeLa cells. One reason might be that TG treatment lasts for only 2 hours, whereas the differential protocol takes days, allowing sufficient time for the global proteome level to change in response to altered protein trafficking dynamics. Nevertheless, since prox-SILAC and pulse-SILAC are looking at different aspects of protein turnover (with prox-SILAC involving also the trafficking aspect), these datasets could complement each other to provide a more wholistic view of protein homeostasis during changes in cellular states.”

“Taken together, the above comparisons highlight the unique advantage of prox-SILAC for mapping spatially resolved protein turnover.”

2) Currently, prox-SILAC method shows less coverage of mitochondrial matrix proteins (~183 proteins) than original report using mito-APEX and SILAC method (~495 proteins) [PMID: 23371551]. It would be great if the authors could clarify this issue in the Discussion part.

Response: We agree with the reviewer that the issue of mito proteome coverage should be discussed. The sample preparation workflows and LC-MS/MS methods are different between the previous mito-APEX experiment and the current work. In the previous paper (PMID: 23371551), following streptavidin bead purification, eluted proteins were loaded on an SDS-PAGE gel which was subsequently cut into 16 gel bands. After in-gel digestion, each fraction was analyzed by LC-MS/MS with LC run time of two hours. In our mitochondrial prox-SILAC experiments, we did not separate the enriched protein on an SDS-PAGE gel, but digested them all on the bead. We speculate that the higher sample complexity have led to lower coverage of mitochondrial proteins. In the revised manuscript, we have added discussions of coverage issue on page 4:

“Notably, the coverage of mitochondrial proteome is lower than the previous report using mito-APEX^[26], which we attribute to differences in the sample preparation workflow and LC-MS/MS methods (see Methods) between two experiments. ”

3) In supplementary methods - Cell culture The exact plasmid name of Delta8.91 should be written. (pCMV-dR8.91?)0.45 uM should be 0.45 um.

Response: Thanks for the suggestion. We have corrected these mistakes.

4) Supplementary table 2,3,4 - Column definitions The turnover rate should be the turnover ratio as the main text. TR should be $H/L / (1 + H/L)$ as the main text.

Response: Thank you for catching this mistake. We have corrected the supplementary tables accordingly.

5) In Fig.1c and 2bpulse-SILAC should be pulse-SILAC.

Response: We apologize for the typo. These have been corrected.

Reviewer #2 (Remarks to the Author):

This manuscript describes a novel application for measuring protein turnover at a subcellular level. While none of the components themselves are new, the combination and resulting information that they obtain is novel and interesting.

Concerns:

How to express and what to call protein turnover is not universally agreed upon. “Turnover”, as expressed here, suggests that the overall level of a protein does not change over the course of the experiment (i.e., assumes steady stage) but I do not

believe that the authors measured the change in expression at their timepoints (4, 8, 12 h). Other variations of turnover involve measuring synthesis and degradation (e.g., in PMID 24045637). Some discussion of this caveat should be added. Or, if the authors have measured total protein changes then their TR values should be expressed relative to this.

Response: We thank the reviewer for raising the issue of the definition of turnover. *Protein turnover* commonly refers to the replacement of older proteins as they are broken down within the cell, whereas our measurement of the prox-SILAC H/L ratio reflects the combined effects of protein synthesis, degradation and local trafficking in specific subcellular localization. Indeed, as the reviewer pointed out, to equate our measured H/(H+L) ratios to protein turnover, we have to assume a steady state of cellular proteome, such as a balanced protein synthesis and degradation. However, cell proliferation should also be accounted for, which suggests that the entire proteome is doubled per cell cycle. At the subcellular level, protein translocation can also influence the SILAC H/L ratio. As these complexities may confuse the readers and users of prox-SILAC technique, we have decided to change the name of *turnover ratio* in the previous submission to *metabolic replacement fraction (MRF)* in the revised manuscript. MRF describes the combined effects of protein synthesis, degradation and translocation on the turn over.

We have added the following discussion to caution the readers of this definition and its difference from conventional protein turnover.

On page 16: “Conventional pulse-SILAC labeling on the whole cell level reveals the protein turnover dynamics, which includes protein synthesis and degradation. Notably, the MRF values measured with prox-SILAC reflects the combine effects of not only protein synthesis and degradation, but also local trafficking in specific subcellular localization.”

The biggest issue with this manuscript is that there is only two replicates of each experiment done and, thus, no statistics to back up some of the biological claims. While I do not like requesting large extra experiments unless truly necessary, in this case I feel that the claims would be very substantially strengthened.

Response: We agree with the reviewer that having only two replicates invites questions regarding the accuracy of H/L measurements. We have carefully analyzed the correlation between our replicated experiments, confirming that all of the Pearson's *r* coefficients between duplicated prox-SILAC experiments are above 0.9, even above 0.95. Thus, we have taken the averaged H/L ratio from duplicated measurements to improve the precision.

While we understand that having three replicates would be nicer, our past experience with our MS facility is that if the runs of replicated samples are separated

by a few months, the overlap in coverage is often low due to variations in the instrument states. Ideally, triplicated samples should be run within a short period of time (e.g. a few days). Thus, all additional MS experiments performed during the revision of this manuscript are in triplicates, but we are hesitant towards repeating all of our experiments in the previous submission, as this could take very long time and resource. We apologize that we are unable to perform the additional replicate as requested by the reviewer, and we hope that our highly correlated duplicated experiments could partially make up for the loss. In the revised manuscript, we have acknowledged the drawback of using duplicates in the Discussion section:

Page 17: “As the MRF values of duplicated prox-SILAC experiments are highly correlated with Pearson’s r coefficients >0.9 , the mean MRF values are used for subsequent analysis.”

How is it possible to have a negative change in the TR value for a given protein from an early timepoint to a later timepoint? This seems to happen for a large fraction of proteins (and is visible in Fig. 1e) but does not seem biologically possible. The authors need to address what could be going on here.

Response: We have also noted the negative changes in MRF, which are not possible biologically. The negative changes often occur when the MRF value is high or almost saturated. Notably, negative changes also exist in the dataset of previous studies with pulse-SILAC but have not been adequately explained (such as PMID: 31974161). For example, the author measured the pulse-SILAC ratios in isolated mitochondria, which revealed that some proteins with fast turnover rates also exhibited higher H/L ratio at 6 hours than 12 hours, e.g. NDUFA4 (1.01 vs. 0.91), TRUB2 (1.87 vs. 1.05) and SPRYD4 (0.67 vs. 0.59). It is likely that the measurement of H/L SILAC ratios is intrinsically noisy. Thus, we tend to attribute the negative changes to measurement errors. In the revised manuscript, we have noted the negative changes both in our study and in the literature to the readers:

Page 17: Notably, negative changes in MRF values during the prox-SILAC time course were observed for some of the proteins, which typically happens when their MRF values are high or almost at saturated levels. Similar trend has also been reported in previous pulse-SILAC studies^[17], which could be attributed to measurement errors of mass spectrometry quantitation. To avoid measurement near the MRF saturation level, we focused on an early timepoint of 2 hr for the subsequent prox- and pulse-SILAC experiments.”

Significant figures in error values. What does it mean that you could measure error to three significant figures? E.g., $56.9 \pm 15.8\%$ on page 4. This is not appropriate and is done throughout the manuscript. The values I have cited here would more

appropriately be expressed as $57 \pm 16\%$.

Response: We apologize for not being careful with significant figures. We have corrected these numbers accordingly.

Reviewer #3 (Remarks to the Author):

Feng Yuan and collaborators combine proximity-dependent protein labeling with enrichment, metabolic labelling and proteomics in cell culture to provide subcellular-level measures of protein turnover.

The technical application of proximity-dependent enrichment and SILAC labelling is relatively little explored and could be potentially useful for this field. The topic of how sub-cellular localization influences protein stability/turnover is extremely interesting.

Unfortunately, I have major concerns about the actual relevance of the work since the authors do not properly showcase the usefulness of their strategy. I see two major purposes for combining proximity-dependent enrichment and SILAC labelling. The first one is that, using this approach, organelle specific sub-proteomes (that cannot be normally detected) become accessible. The second one is that looking at the subcellular proteome vs. the total proteome some important changes can be observed in the labelling efficiency. The main problem of this manuscript is that the authors consider only very superficially these two aspects.

To address these major concerns, the authors should systematically quantify protein lifetimes/labelling profiles across sub cellular proteomes, compare them to the total proteome, report proteins that cannot be seen in the overall proteome and show that indeed the localization of proteins can influence the relative age/labelling profile of sub cellular proteomes.

Major

1) Fig. S1: which should be the enriched fraction? The eluted sample? From panel A it does not seem to be the case and it looks like the enrichment did not work at 4h. Could the authors comment on that?

Response: We thank the reviewer for the careful examination. Indeed, in panel A of Fig S1, the high biotinylation signal in the supernatant (S fraction) indicates quite low enrichment efficiency for the 4 hr experiment. We also noticed that the number of proteins captured in this experiment (229 proteins) is much lower than those at 8 hr

(612) and 12 hr (410), which is likely due to the lower enrichment efficiency. However, data in Table S1 show that the mitochondrial specificity for the 4 hr experiment is actually quite high (85%), which suggests that proteins specifically labeled by APEX2 are indeed enriched, although only at a low level.

As outlined in the workflow of panel D in Fig S1, the majority of biotinylated proteins binding to the beads are digested and released into the supernatant in the form of tryptic peptides. Before on-bead digestion, only a small fraction of beads was retained for the “elution” sample preparation and gel analysis (E fraction). Thus, the amount of proteins shown in the E fraction of gel does not represent the actual amount of tryptic peptide derived from the beads. It is likely that, despite low enrichment efficiency for the 4 hr experiment, the small amount of digested peptide could still be analyzed by LC/MS-MS, which has high sensitivity. In terms of SILAC quantitation, as both heavy and light forms of the same peptide are enriched at identical efficiency, the level of recovery does not affect the H/L ratio. Thus, only the coverage is negatively affected by the low efficiency.

We realized that the original statement of “both western blot analysis and silver staining confirmed successful biotinylation and protein enrichment (Figure S1)” is not accurate. In the revised manuscript, we have noted the low enrichment level at 4 hr to the readers (page 3):

“Both western blot analysis and silver staining confirmed successful biotinylation and protein enrichment across replicates at 8 hr and 12 hr (Figure S1).”

“For the 4 hr experiment, silver staining revealed low protein enrichment efficiency, but sufficient amount of peptides were obtained for subsequent MS analysis (Figure S1).”

Figure S1. Immunoblotting and silver staining characterization of Prox-SILAC labeling in HEK293T mito-APEX2 cell line. (D) Scheme for WB sample preparation.

2) A quick GO over-representation analysis of the 162 proteins that were identified and quantified across different replicates (end of page 3) reveals that beside the mitochondrial matrix there are several proteins that belong to the nucleoid and to other compartments. This raises the question of how precise is the enrichment method applied. This is particularly relevant in the light of what the authors write in their introduction: "However, organelle purification methods are prone to introducing contamination and are not generally applicable to other sub-cellular locations, which motivates us to explore alternative methods for studying sub-cellular proteome turnover." How much quantitatively enriched are mitochondrial proteins using their approach vs other proteins? A measure of relative enrichment vs initial abundance should be implemented in the analysis to facilitate the interpretation of the results. If possible, a comparison with other sub cellular fractionation methods should also be implemented, to showcase the usefulness of the approach proposed here. One other angle could be to showcase proteins that are not detected by normal MS in the pulse SILAC but that can be observed following enrichment (and thus deeper coverage of that sub cellular proteome). Similar to the mitochondrial labelling, the other enrichment strategies should be compared to other available fractionation methods.

Response: We thank the reviewer for the advice on discussing the unique advantages of implementing proximity-labeling. We have double-checked our list of 162 mitochondrial proteins identified in three replicates (Fig.1C). Regarding the "nucleoid" GOCC term, the reviewer might be referring to Hsp40, which has been shown to localize to the nucleoid (PMID: 23688635). However, we note that the term "nucleoid" here actually refers to "mitochondrial nucleoid", which describes the complex formed between mitochondrial DNA and proteins in the mitochondrial matrix. Thus, this "nucleoid" GOCC term does not contradict with mitochondrial specificity. Similarly, the "ribosome" GOCC term of MRPL21 actually refers the "mitochondrial ribosome", which is also localized in the matrix.

It is possible that some of the mitochondrial proteins may have multiple subcellular localizations, leading to the inclusion of other GOCC terms in the list. To help the readers evaluate the mitochondrial specificity of our dataset, we have supplemented the GOCC analysis of the 162 overlapped proteins in Figure S4B. The most over-represented GOCC terms are "mitochondrion", "mitochondrial matrix", "mitochondrial inner membrane" and "mitochondrial nucleoid".

Regarding the relative enrichment level, as we have used SILAC to measure protein turnover, we need to apply other means of isotope labeling to quantitatively evaluate enrichment levels. We have attempted to combine pulse-SILAC with peptide demethylation labeling, but the presence of both heavy amino acid and heavy methyl has substantially increased the complexity of MS data, which dramatically reduced the

number of identified proteins. Based on our past quantitative proteomic experiments, APEX2-mediated proximity labeling could achieve $\log_2(H/L) > 7$ for the top 200 enriched proteins (PMID: 32470320). In the current study, we sought to improve the mitochondrial specificity by taking the overlap of multiple replicates across multiple time points. In the case of mitochondrial matrix, a total of six datasets are used to derive our list of mitochondrial proteins (Figure S3).

In terms of protein coverage, we thank the reviewer for suggesting us “to showcase proteins that are not detected by normal MS in the pulse SILAC but that can be observed following enrichment”. In the revised manuscript, we have provided such analysis. In a previous work, Bogenhagen and co-workers performed pulse-SILAC experiment on HeLa cell for 3-12 hrs and subsequently purified mitochondria (PMID: 31974161). They quantified the H/L SILAC ratios of a total of 669 proteins, which are distributed across four sub-mitochondrial compartments: matrix, inner membrane, inter-membrane space, and outer membrane. In comparison, our mitomatrix-targeted APEX2 almost exclusively labels matrix proteins and matrix-exposed IMM proteins, thus offering higher sub-mitochondrial specificity (Figure S4A). 33 proteins are uniquely identified in our mitomatrix prox-SILAC but not in the Bogenhagen paper. To directly compare the coverage of mitochondrial proteome using the same cell lines, in the revised manuscript, we have performed additional pulse-SILAC experiments and compared our mitomatrix prox-SILAC against our own whole-cell pulse-SILAC x mito annotation analysis, revealing that 72 proteins are uniquely identified in the mitomatrix prox-SILAC dataset. In the revised manuscript, we have provided the above comparisons:

Page 18: “For example, our mitochondrial prox-SILAC dataset contains 72 proteins that are missed by mito pulse-SILAC, despite the overall larger coverage of the latter (407 mito proteins vs. 162).”

We have extended the above analysis to the ER lumen. To make a fair comparison, we have performed additional pulse-SILAC experiments in the relevant cell lines. In HEK293T cells stably expressing HRP-KDEL, 94 proteins in our ER-prox-SILAC dataset are uniquely identified in prox-SILAC (i.e. not in our own pulse-SILAC x ER annotation). In HeLa-SS-HRP-KDEL cells, 172 proteins and 186 proteins were identified and quantified only in prox-SILAC experiments, in the presence and absence of TG treatment, respectively. During the differentiation of SH-SY5Y cells, 262, 180 and 90 proteins are only identified and quantified in prox-SILAC experiments at D0, D7 and D10 state, respectively.

In the revised manuscript, we have provided the above comparisons:

Page 18: “Similarly, our ER-lumen prox-SILAC datasets in HEK293T cells, HeLa cells, and TG-stressed HeLa cells have uniquely identified 94, 186, and 172 proteins, respectively, as compared to the pulse-SILAC counterparts. With our newly added

pulse-SILAC datasets for SH-SY5Y cell differentiation, we confirm that 262, 180, and 90 proteins are uniquely quantified in ER lumen prox-SILAC at D0, D7, and D10, respectively, as compared to the corresponding pulse-SILAC datasets.”

Finally, we note that organelle purification is not a general solution to acquire subcellular proteomes. Unlike the whole mitochondria, it is extremely challenging, if not impossible, to purify the intact ER, the outer mitochondrial membrane, or the outer ER membrane. For example, microsome preparation is often used as a proxy to the ER. In contrast, proximity labeling has provided a generally applicable approach to access the proteome in these subcellular compartments. We briefly discussed this issue in the revised manuscript, to highlight the advantage of prox-SILAC.

Page 16: “Isolated microsome was used to profile ER proteome, but generating incomplete datasets with poor overlap in independent works^{[51,52]”}.

Figure S4. Specificity analysis of our mito-prox-SILAC dataset. (A) Sub-mitochondrial specificity analysis of our mito-prox-SILAC dataset, comparing with MitoCarta3.0 database and previous work (pulse-SILAC with mitochondria isolation) ^[2]. (B) GOCC analysis of the 162 proteins identified in our mitochondrial dataset with previous mitochondrial annotations. The numbers of proteins with corresponding GOCC annotations are list on the left of the chart.

3) The turnover ratio (%) used across the manuscript (introduced in the upper part of Page 4), is in this work a measure of the metabolic labelling. When express to a percentage actually corresponds to the Heavy fraction (%). To avoid misunderstandings, since proteins generally get labeled with exponential dynamics, this measure should be renamed to "normalized heavy fraction" since a measure of turnover in this field often indicates a certain measure of time (a turnover rate) and the term could be misleading. Have the authors considered to measure the half-life (the lifetimes) for the proteins analyzed?

Response: We agree with the reviewer that our definition of protein turnover based on normalized H/L ratio might mislead the readers. Indeed, *protein turnover* often refers

to the replacement of older proteins as they are broken down within the cell, whereas our measurement of the prox-SILAC H/L ratio reflects the combined effects of protein synthesis, degradation, and local trafficking in specific subcellular localization. Thus, in the revised manuscript, we have changed the name of *turnover ratio* to *metabolic replacement fraction (MRF)*. This new term is essentially the same as “normalized heavy fraction” as suggested by the reviewer. We have also added the following discussion to caution the readers of this definition and its difference from conventional protein turnover.

On page 16: “Conventional pulse-SILAC labeling on the whole cell level reveals the protein turnover dynamics, which includes protein synthesis and degradation. Notably, the MRF values measured with prox-SILAC reflects the combine effects of not only protein synthesis and degradation, but also local trafficking in specific subcellular localization.”

As suggested by the reviewer, we have also calculated the $t_{1/2}$ of the ER proteins quantified across all the time points (1h/2h/4h/8h/12h). For the overlapped 183 proteins, the $t_{1/2}$ is broadly distributed from 0.9 h to 24.7 h with a mean value of 14.3 h. Proteins with long half-lives are mainly ER resident proteins, such as calreticulin (24.7 h) and malectin (20.9 h). Proteins with short half-life time in ER are mainly secreted proteins, such cell surface receptor APP (0.9 h) and TFR1 (1.0 h). We have supplemented these results on page 9:

“In addition, we calculated the $t_{1/2}$ of the 183 proteins identified in all five time points. Their $t_{1/2}$ exhibited a distribution from 0.9 h to 24.7 h with a mean $t_{1/2}$ of 14.3 h (Table S2, Figure S15). The protein with the shortest $t_{1/2}$ (0.9 h) in the ER is amyloid-beta precursor protein (APP), a cell surface receptor relevant to neurite growth and neuronal adhesion. On the contrary, the protein with the longest $t_{1/2}$ (24.7 h) is calreticulin, an ER resident protein.”

Figure S15. The distribution of half-life times of 183 ER proteins identified in 1h/2h/4h/8h/12h prox-SILAC experiments.

4) It is obvious that longer pulses would correspond to higher metabolic labelling. Likewise, differences in protein lifetimes in mitochondrial proteins (and in mitochondrial ribosomes) have been previously observed with no need of proximity labelling enrichment. The relevant question here is what is the difference in the metabolic labelling of the mitochondrially-enriched proteins vs. the labelling of non-enriched proteins. Do the authors see a difference? This aspect should be addressed systematically taking into account all labelling time points and enriched vs. not-enriched proteomes. As also mentioned on point 1, one other possibility, if there are no differences would be to show that the proximity labelling has other advantages (such as deeper coverage of the proteome). The advantages of this approach should be more clearly presented in the work.

Response: We thank the reviewer for the suggestion of analyzing differences between enriched and non-enriched proteins, which could be used to highlight the advantages of proximity labeling. In the revised manuscript, we have performed additional pulse-SILAC experiments in HEK293T-mito-APEX2 cell lines with 8 hours metabolic incorporation. We compared our prox-SILAC dataset against the pulse-SILAC dataset, which represents non-enriched proteins. For the 111 proteins identified in both prox-SILAC and pulse-SILAC datasets, their overall MRF_{8h} values are comparable between the two datasets. The notable exceptions are some OXPHOS subunits (e.g. NDUF9, 76±7% vs. 27±7%; ATP5C1, 69±1% vs. 32±1%). The mean MRF_{8h} value of OXPHOS subunits in pulse-SILAC (34%) is lower than that in prox-SILAC (50%), a difference that may be attributed to protein translocation. As subunits in the OXPHOS machinery are liable to be oxidized and need to be replaced, excessive copies in the mitochondria could act as a reserve that is necessary to keep the OXPHOS machinery functioning, thus leading to higher turnover for a subset of mitochondrial proteins. Our data suggest that a small sub-population of NDUF9 and ATP5C1 may not be accessible to mitomatrix-targeted APEX (e.g. on the outer membrane as newly synthesized protein, or outside mitochondria), although we do not have experimental data to support this view. The protein over-synthesis and higher heavy fraction in mitochondria phenomenon was observed in previous work (PMID: 31974161).

Regarding the reviewer's suggestion of investigating whether prox-SILAC offers "deeper coverage of the proteome", please see our response to issue #2. Taken together, all above evidence proved the necessity of proximity labeling on investigating mitochondrial protein turnover. We have supplemented the above results in the revised manuscript:

Page 5: "We then performed pulse-SILAC experiments with mito-APEX2 cell line to evaluate the MRF values at the whole cell level. From three replicated 8-hour pulse-SILAC experiments, we identified and quantified 1584 proteins from at least two

replicates (Figure S8A and S9, Table S1). For the 111 proteins identified in both prox-SILAC and pulse-SILAC datasets (Figure S8B), their overall MRF_{8h} values are similar between the two datasets (Figure S10). The notable exceptions are several OXPHOS subunits: NDUFB9 (76±7% vs. 27±7%), ATP5C1 (69±1% vs. 32±1%) and COX5A (56±2% vs. 34±2%) (Figure S10). The pulse-SILAC MRF_{8h} values of OXPHOS components range from 26% to 56% (mean MRF_{8h} value 34%), which is lower than the corresponding prox-SILAC MRF_{8h} values (from 24% to 78%, mean value 50%).”

Page 16: “Our data suggest that a small sub-population of NDUFB9 and ATP5C1 may not be accessible to mitomatrix-targeted APEX (e.g. on the outer membrane as newly synthesized protein, or outside mitochondria), although we do not have experimental data to support this view.”

Figure S8. Venn diagrams showing the numbers of quantified proteins in HEK293T-mito-APEX2 cells from three replicated experiments. (A) Venn diagram showing the overlap of three replicates. (B) Overlap of 111 proteins identified both in pulse-SILAC and prox-SILAC.

Figure S10. Scatter plot of MRF_{8h} values of mitochondrial proteins measured in mito-prox-SILAC against whole-cell pulse-SILAC. The zoom-in view with error bars is shown on the right.

5) Although it is tantalizing to speculate that the secretory pathway shows a specific clearance of older proteins that would be trafficked to the Golgi (end of page 6), it

cannot be excluded that faster labelling in the ER simply reflects an overall faster turnover of all ER proteins vs mitochondrial proteins (as previously reported in several occasions in vitro and in vivo). The point of how sub-cellular localization influences protein turnover should be the most important aspect of this work. In order to address this the authors should do proximity labelling in different organelles and compare the labeling across sub-proteomes and with the overall proteome (see below). In this context, what is the definition of secreted proteins for the authors? Proteins exposed to the plasma membrane or actually "strictly secreted" (e.g. hormones) and/or cleaved proteins such as APP? Strictly secreted proteins are lost in the medium upon secretion and proteolytic cleavage and their labelling profile will reflect this process. In other words, their lifetimes will be extremely short (so higher relative H labeling as in reality the only thing that is measured is how long they stay within the cell, not how long they actually live). This observation, has not much to do with the fact that proteins trafficked to the Golgi are having a shorter lifetime. Moreover, the difference might be explained by evolutionary constraints that render some proteins more or less prone to damage or degradation. For this type of analysis proteins from the plasma membrane (that are not secreted) could be taken as a proxy and their relative labeling could be measured in the ER, in the Golgi and in the plasma membrane upon sub-cellular fractionation or surface labelling or, if the authors wish, using complementary proximity labelling approaches. The authors should address comparatively if at the same labelling time point the protein localization influences the labelling efficiency. The authors should refrain from using a scheme in support of their hypothesis.

Response: We thank the reviewer for the suggestion of measuring MRF values in different subcellular compartments along the secretory pathway. In this study, secreted proteins refer to both strictly secreted soluble proteins (i.e. secreted outside the cells, like hormones) and membrane proteins residing on the plasma membrane, Golgi or lysosome (i.e. post-ER compartments in the secretory pathway). We realize that our definition of "secreted/secretory protein" is causing confusion to the reader. Therefore, in the revised manuscript, we have changed the name to "post-ER secretory pathway proteins" to more accurately describe the subcellular localization of these proteins.

To investigate the dynamic protein turnover on the plasma membrane, we have performed additional pulse-SILAC experiments and labeled the cell surface protein population with the membrane-impermeant chemical reagent, Sulfo-NHS-LC-Biotin. Similar to our workflow of prox-SILAC, biotinylated proteins were subsequently enriched with streptavidin-coated beads (Figure S18-S19). For the 34 proteins identified in both ER lumen and cell surface prox-SILAC datasets, their MRF values differ substantially (Figure S20). Membrane receptors and ion pumps such as TFRC (71 ± 1 vs. 4 ± 0), ATP1A1 (68 ± 1 vs. 19 ± 0) and IGF2R (55 ± 3 vs. 5 ± 3), exhibit much higher ER lumen MRF values, which is attributed to the time required for them to traffic

from the ER lumen to the plasma membrane. On the contrary, several protein chaperones (e.g. PDIA6) showed lower MRF values in the ER than on the cell surface. While normally residing in the ER, PDIA6 has also been reported to associate with other substrate on the cell surface (PMID: 25624318). We thus speculate that the higher MRF values of protein chaperones may be associated with their distinct functions on the cell surface. For the few protein examples with higher MRF values on the plasma membrane than in the ER lumen, it seems that these newly synthesized proteins are preferentially targeted to the plasma membrane before subsequently returning to the ER. However, experimental proof of this unusual trafficking pattern is currently lacking and may stimulate future research on this topic.

We also agree with the reviewer that it would be very informative include Golgi to the prox-SILAC dataset comparisons. However, for technical reasons, proximity labeling in the Golgi apparatus has not been possible, despite numerous attempts over the past decade. An on-going project in our laboratory has recently solved the problem of proximity labeling in cis-Golgi and trans-Golgi. However, this is quite beyond the scope of the current study. In the revised manuscript, we have added the above results and discussions.

Page 9: “To further investigate protein turnover on the plasma membrane, we performed 2-hour pulse-SILAC labeling in HEK293T cells and labeled the cell surface protein population with the membrane-impermeant chemical reagent, Sulfo-NHS-LC-Biotin. Similar to our workflow of prox-SILAC, biotinylated proteins were subsequently enriched with streptavidin-coated beads and analyzed by LC-MS/MS to derive the MRF_{2h} values (Figure S18). In two replicated experiments, we have identified and quantified 508 cell surface proteins (Figure S19). For the 34 proteins identified in both ER lumen and cell surface prox-SILAC datasets, their MRF_{2h} values differ substantially (Figure S20). Membrane receptors and ion pumps such as TFRC (71±1% vs. 4±0%), ATP1A1 (68±1% vs. 19±0%) and IGF2R (55±3% vs. 5±3%), exhibit much higher ER lumen MRF values, which is attributed to the time required for them to traffic from the ER lumen to the plasma membrane. To our surprise, two protein chaperones, SERPINH1 (5.0±0% vs. 86±0%) and PDIA6 (4±5% vs. 56±16%) exhibit higher cell surface MRF_{2h} than ER MRF_{2h} values. SERPINH1 (Hsp47) is a collagen-specific molecular chaperone and plays an important role in the collagen biosynthesis^[35]. While PDIA6 mainly functions as a chaperone that inhibits aggregation of misfolded proteins during unfolded protein response (UPR)^[36], it has also been reported to bind integrin β3 subunit on the cell surface to promote platelet activation after stimulation^[37]. We thus speculate that the higher MRF values of protein chaperones may be associated with their distinct functions on the cell surface. Together, the above comparison of prox-SILAC at various subcellular compartments highlight its advantage in investigating spatially resolved protein turnover.”

Figure S18. Experimental scheme of pulse-SILAC in HEK293T-SS-HRP-KDEL cells and cell surface protein enrichment. To tag the nascent proteins, HEK293T-SS-HRP-KDEL cells were briefly cultured in heavy SILAC medium for 2 hours. To label cell surface proteome, cells were incubated with sulfo-NHS-LC-biotin for 10 mins to capture the cell surface proteins. The nascent PM proteins were thus doubly labeled with both biotin and heavy arginine/lysine. Following cell lysis, biotinylated proteins were enriched by streptavidin-coated beads. The metabolic replacement fractions of targeted proteins were identified by subsequent LC-MS/MS analysis.

Figure S19. Pulse-SILAC experiments on cell surface. (A) Venn diagram showing the overlap of the two cell surface pulse-SILAC replicates. (B) Scatter plot showing the MRF values of the two replicates.

Figure S20. Scatter plot of MRF_{2h} values of secretory pathway proteins measured in ER-prox-SILAC against cell surface pulse-SILAC.

6) Are changes observed in Fig. 3E and Fig. 4E specific for the ER-enriched proteome or these changes are general (and could be thus observed also in the absence of proximity labelling and enrichment)? The authors should address this point and showcase the usefulness of their method and why looking at this sub-cellular proteome is relevant for studying these issues.

Response: We agree with the reviewer that it would be important to perform a head-to-head comparison of prox-SILAC against pulse-SILAC. In the revised manuscript, we have performed the corresponding pulse-SILAC version of experiments shown in the original Figures 3E and 4E. Specifically, in HeLa-SS-HRP-KDEL cell line, we performed pulse-SILAC in the presence and absence of TG treatment; in SH-SY5Y-SS-HRP-KDEL cell line, we measured pulse-SILAC at D0, D7 and D10 of chemically stimulated neurite growth. Notably, we demonstrate that changes observed in Figures 3E and 4E are most prominent in prox-SILAC but not in pulse-SILAC x subcellular information.

In revised Figure 3E, we found that ER stress caused a significant global reduction of MRF_{2h} (TG-treated vs. control) measured in prox-SILAC (downward deviation from the diagonal) but to much less extent in pulse-SILAC. This indicates that TG treatment is causing substantial changes in the landscape of newly synthesized proteins within 2 hours (mean values 18% vs 26%), but impact has barely reached the global proteome level (i.e. protein abundances) (mean values 13% vs 14%). Even after we extracted the secretory pathway proteins from the pulse-SILAC experiments using GOCC annotations, generating the pulse-SILAC x subcellular information dataset, the global reduction in MRF_{2h} was still minimal (mean values 13% vs 14%) (Figure 3E).

Similarly, in Figure 4E, most proteins exhibit decreasing MRF_{2h} values from D7 to D10 (downward deviation from the diagonal), but the trend is less noticeable in the corresponding pulse-SILAC dataset or the pulse-SILAC x subcellular information dataset (Figure S31). However, we note that the difference between prox-SILAC and pulse-SILAC is less striking than in the case of TG treatment in HeLa cells. One reason might be that TG treatment lasts for only 2 hours, whereas the differential protocol takes days, allowing sufficient time for the global proteome level to change in response to altered protein trafficking dynamics. Nevertheless, since prox-SILAC and pulse-SILAC are looking at different aspects of protein turnover (with prox-SILAC focusing also on the trafficking aspect), these datasets could complement each other to provide a more holistic view of protein homeostasis during cellular differentiation. In the revised manuscript, we have added the above results and discussion.

Page 12: “We also performed 2 hours pulse-SILAC experiments using HeLa-SS-HRP-KDEL cell line with or without TG treatment. We have identified and quantified the MRF_{2h} values of 1429 proteins in three TG treatment replicates and 1598 proteins

in three control replicates, resulting in 1202 overlapped proteins (Figure S23-S24, Table S3). Overall, much higher MRF_{2h} values were observed for post-ER secretory pathway proteins in the prox-SILAC experiments (Figure S25). While ER stress causes a significant global reduction of MRF_{2h} (TG-treated vs. control) measured in prox-SILAC (mean values 18% vs 26%), the decrease is much less in pulse-SILAC (mean values 13% vs 14%). Even after we extracted the secretory pathway proteins from the pulse-SILAC experiments using GOCC annotations, generating the pulse-SILAC x subcellular information dataset of 584 proteins, the global reduction in MRF_{2h} was still minimal (mean values 13% vs 14%) (Figure 3E)."

Page 18: "The observed change in ER prox-SILAC MRF_{2h} values during TG treatment may have been eclipsed by the whole-cell pulse-SILAC MRF_{2h} values. For example, two heat shock proteins, HSPA5 (from 4±0% to 15±1%) and HSPA6 (from 4±0% to 16±0%), have more than 3-fold changes in ER prox-SILAC MRF_{2h} but their corresponding pulse-SILAC MRF_{2h} increased only slightly upon TG treatment (HSPA5: from 6±5% to 9±5%; HSPA6: from 3±1% to 6±2%) (Figure 3E). While TG treatment is causing substantial changes in the landscape of newly synthesized proteins within 2 hours, the impact has barely reached the global proteome level (i.e. protein abundances)."

Page 15: "In parallel, we performed 2-hour pulse-SILAC experiments at Days 0, 7, 10 during neurite growth, resulting in the identification of 1687, 1381 and 1496 proteins at these time points, respectively (Figure S28-S29, Table S4). Similar to our previous comparisons between prox-SILAC and pulse-SILAC in the ER, overall higher MRF_{2h} values are observed in prox-SILAC at D0 and D7, particular for post-ER secretory pathway proteins (Figure S30). However, the difference is much smaller at D10, where the MRF_{2h} values are lower in both prox-SILAC and pulse-SILAC (Figure S30)."

"During the early stage of differentiation (from D0 to D7), protein neudesin was identified with 5-fold increase of MRF_{2h} in prox-SILAC experiments (Figure 4D), but the MRF_{2h} showed nearly 2-fold decrease (from 29±16% to 16±7%) in pulse-SILAC experiments (Figure S31). Prolyl 3-hydroxylase 1 and N-CAM-1 also exhibited opposite MRF_{2h} changes between prox-SILAC and pulse-SILAC (Table S4). Between D7 and D10, while most proteins exhibit decreasing MRF_{2h} values from D7 to D10 (downward deviation from the diagonal), the trend is less noticeable in the corresponding pulse-SILAC dataset or the pulse-SILAC x subcellular information dataset (Figure S31)."

Page 18: "However, we note that the difference between prox-SILAC and pulse-SILAC is less striking in SH-SY5Y differentiation experiment than in the case of TG-induced ER stress in HeLa cells. One reason might be that TG treatment lasts for only 2 hours, whereas the differential protocol takes days, allowing sufficient time for the global proteome level to change in response to altered protein trafficking dynamics.

Nevertheless, since prox-SILAC and pulse-SILAC are looking at different aspects of protein turnover (with prox-SILAC involving also the trafficking aspect), these datasets could complement each other to provide a more wholistic view of protein homeostasis during changes in cellular states.”

Figure S29. Venn diagrams showing the overlap of quantified proteins in 2h pulse-SILAC experiments and prox-SILAC experiments during the differentiation of SH-SY5Y-SS-HRP-KDEL cells.

Figure S30. Scatter plot of MRF_{2h} values of ER proteins measured in ER prox-SILAC against whole-cell pulse-SILAC in SH-SY5Y-SS-HRP-KDEL cells during differentiation. Red dots indicate post-ER proteins (i.e. cell membrane, Golgi apparatus, and lysosome), while blue dots indicate ER resident proteins.

Figure 4. Investigating the turnover changes of ER proteins during neurite growth. (E). Red and blue dots represent proteins with 2-fold higher and lower MRF_{2h}, respectively.

Figure S31. Scatter plot comparing the MRF_{2h} values in pulse-SILAC experiments at different states during cell differentiation. (A) left: MRF values of the whole cell proteins identified at both D0 and D7. Right: MRF values of proteins with secretome annotations identified at both D0 and D7. (B) left: MRF values of the whole cell proteins identified at both D7 and D10. Right: MRF values of proteins with secretome annotations identified at both D7 and D10. Red and blue dots represent proteins with 2-fold higher or lower MRF_{2h} values, respectively.

Minor

0) For making comments on specific text sections it would be really useful to have line numbers.

Response: We thank the reviewer for the suggestion and have added the line numbers in our revised manuscript.

1) The introduction does focus predominantly on *in vitro* cellular work, although great progress has been made in the protein turnover field *in-vivo* (in whole animals). Since the analysis of protein lifetimes in culture might reflect a very limited representation of the situation *in vivo*, the authors should at least acknowledge that literature that literature in the introduction.

Response: We thank the reviewer for suggestion. We have cited the literature about protein turnover mapping in mouse tissues in the revised Introduction section.

On page 2: “More recently, stable isotope labels (pulse-SILAC^[12,13]) have been used to quantify protein turnover in mammalian cell lines, primary culture, such as neurons^[6] and *in vivo* system^[14,15].”

2) There is at least an exception to the statement “While these methods have been broadly applied to profile the abundance of proteins within subcellular structures they have not been used for investigating the turnover dynamics of subcellular proteome.”. There is one work using MIMS (NanoSIMS) combining APEX and metabolic labelling: <https://pubmed.ncbi.nlm.nih.gov/31719114/> which should be probably cited in this context. The authors should also check more carefully if there are other such

examples in the literature and acknowledge them.

Response: We thank the reviewer for the suggestion. We have revised the Introduction to include the proper citations of previous work.

On page 3: “In a previous work, APEX labeling was coupled with mass spectrometry imaging and metabolic labeling to reveal the protein turnover in lysosome^[30]. However, the work has not been extended to proteome level.”

3) The quality of Fig s15 I not acceptable (the image is completely blurred / out of focus).

Response: We thank the reviewer for the suggestion and have updated the supplementary figure S15.

Figure S26. Immunofluorescence characterization of the localization of HRP-eGFP-KDEL in SH-SY5Y cells. The expression of HRP-eGFP-KDEL, ER marker calnexin and biotinylated proteins (streptavidin-AlexaFluor 647) are merged. Scale bar, 10 μm .

REVIEWERS' COMMENTS

Reviewer #1 (Remarks to the Author):

In the revised manuscript, the authors have conducted additional experiments, such as cell surface biotinylation, and have compared these results with the HRP-KDEL (ER lumen) prox-SILAC experiment. With the inclusion of the additoinal results and revised term "metabolic replacement fraction," the manuscript has been nicely improved, effectively emphasizing the utility of the prox-SILAC method and the noteworthy biological discoveries presented in this paper. Consequently, this reviewer strongly recommends accepting this manuscript for publication in Nature Communications without any further need for revisions.

Reviewer #2 (Remarks to the Author):

The authors have adequately addressed my concerns

Reviewer #3 (Remarks to the Author):

I genuinely appreciate the authors' dedicated efforts in addressing all comments. While some attempts didn't yield the desired results, I acknowledge the substantial progress made by the authors and believe that their work merits publication.

Minor:

I kindly request the addition of a brief paragraph at the end of the discussion that highlights potential limitations of the APEX approach. For instance, there seems to be a bias in APEX labeling towards newer proteins within the mitochondrion, possibly due to increased accessibility of those vs. the aged one (as probably hinted by the authors in lines 442-444). This aspect, along with other potential APEX limitations such as pH and redox APEX sensitivity, warrants careful consideration within the research community when interpreting these findings and should be stated clear. Despite the increased complexity in result interpretation, the authors can frame this positively by emphasizing that APEX labeling still offers valuable complementary data with discovery potential.

REVIEWERS' COMMENTS

Reviewer #1 (Remarks to the Author):

In the revised manuscript, the authors have conducted additional experiments, such as cell surface biotinylation, and have compared these results with the HRP-KDEL (ER lumen) prox-SILAC experiment. With the inclusion of the additional results and revised term "metabolic replacement fraction," the manuscript has been nicely improved, effectively emphasizing the utility of the prox-SILAC method and the noteworthy biological discoveries presented in this paper. Consequently, this reviewer strongly recommends accepting this manuscript for publication in Nature Communications without any further need for revisions.

We thank the reviewer for these positive remarks.

Reviewer #2 (Remarks to the Author):

The authors have adequately addressed my concerns

We thank the reviewer for helping us improve the manuscript.

Reviewer #3 (Remarks to the Author):

I genuinely appreciate the authors' dedicated efforts in addressing all comments. While some attempts didn't yield the desired results, I acknowledge the substantial progress made by the authors and believe that their work merits publication.

Minor:

I kindly request the addition of a brief paragraph at the end of the discussion that highlights potential limitations of the APEX approach. For instance, there seems to be a bias in APEX labeling towards newer proteins within the mitochondrion, possibly due to increased accessibility of those vs. the aged one (as probably hinted by the authors in lines 442-444). This aspect, along with other potential APEX limitations such as pH and redox APEX sensitivity, warrants careful consideration within the research community when interpreting these findings and should be stated clear. Despite the increased complexity in result interpretation, the authors can frame this positively by emphasizing that APEX labeling still offers valuable complementary data with discovery potential.

We appreciate the reviewer's advice. In the revised manuscript, we have added a paragraph in the Discussion section to highlight potential limitations of prox-SILAC. As the reviewer pointed out, APEX labeling appears biased towards newly synthesized

proteins and the efficiency may be affected by local pH and redox states. Such bias should be considered when interpreting prox-SILAC data.

Page 18: “It is noteworthy that, in addition to their differences in the spatial resolution (subcellular vs. whole-cell), prox-SILAC and pulse-SILAC are also looking at different aspects of protein turnover. For example, unlike pulse-SILAC, prox-SILAC is affected by subcellular protein trafficking. In addition, our data have indicated a bias in APEX2 labeling towards newer proteins within the mitochondrion, possibly due to their increased accessibility as compared to their older counterparts. This aspect, along with other potential APEX2 limitations such as sensitivity towards local pH and redox states, warrants careful considerations when interpreting prox-SILAC data. Nevertheless, despite the increased complexity in result interpretation, prox-SILAC still offers valuable complementary data to pulse-SILAC with discovery potentials.”